# Bacteria from the *Amycolatopsis* genus associated with a toxic bird secrete protective secondary metabolites

Elena Seibel[1,2,12], Soohyun Um[2,3,12], Kasun H. Bodawatta[4], Anna J. Komor[5], Tanya Decker[1], Janis Fricke [1], Robert Murphy [6], Gibson Maiah[7], Bulisa Iova[8], Hannah Maus[9], Tanja Schirmeister[9], Knud Andreas Jønsson [4,10], Michael Poulsen [6] & Christine Beemelmanns [1,2,11] ✉

Uropygial gland secretions of birds consist of host and bacteria derived compounds and play a major sanitary and feather-protective role. Here we report on our microbiome studies of the New Guinean toxic bird *Pachycephala schlegelii* and the isolation of a member of the *Amycolatopsis* genus from the uropygial gland secretions. Bioactivity studies in combination with co-cultures, MALDI imaging and HR-MS/MS-based network analyses unveil the basis of its activity against keratinolytic bacteria and fungal skin pathogens. We trace the protective antimicrobial activity of *Amycolatopsis* sp. PS_44_ISF1 to the production of rifamycin congeners, ciromicin A and of two yet unreported compound families. We perform NMR and HR-MS/MS studies to determine the relative structures of six members belonging to a yet unreported lipopeptide family of pachycephalamides and of one representative of the demiguisins, a new hexapeptide family. We then use a combination of phylogenomic, transcriptomic and knock-out studies to identify the underlying biosynthetic gene clusters responsible for the production of pachycephalamides and demiguisins. Our metabolomics data allow us to map molecular ion features of the identified metabolites in extracts of *P. schlegelii* feathers, verifying their presence in the ecological setting where they exert their presumed active role for hosts. Our study shows that members of the Actinomycetota may play a role in avian feather protection.

The chemical ecology of bird–bacteria interactions has received considerable attention over past decades as wild birds act as reservoirs for emerging zoonotic bacterial pathogens[1]. Birds have evolved a variety of protective measures against predators and microbial pathogens, including sanitary behaviors, toxins, gland secretions, and presumed protective microbial symbionts[2–5]. In particular, the uropygial gland (UG), a holocrine secretory gland situated at the dorsal base of a bird's tail, contributes to waterproofing and safeguarding the plumage and feathers against microbial pathogens and ectoparasites; mainly through the secretion of an odoriferous, antimicrobial preen oil that is

a mixture of waxes and volatiles[6–12]. Studies analyzing the bacterial communities and their possible contributions within UGs and feathers across bird species have documented pronounced prevalence of γ- and α-Pseudomonadota, occasionally accompanied by Actinomycetota, as well as keratinolytic feather-degrading bacteria (FDB), such as *Bacillus licheniformis* or *Kocuria rhizophila*, that can negatively affect feather integrity and thus likely avian health[3,4,11–14]. Based on co-cultures, it was anecdotally hypothesized that the avian microbiome could encompass protective bacterial species to counteract activities of FDB. In one example, *Enterococcus faecalis* isolates from the

Eurasian hoopoe preen oil was indeed inhibitory against FDB due to the secretion of bacteriocins (enterocin MR10)[8,15].

However, despite the growing body of evidence underscoring the importance of bacterial symbionts in avian health, chemical evidence remains scarce. To address this knowledge gap, and to explore whether bacterial symbionts could contribute to the protection against FDB by chemical means, we conducted a comprehensive chemical-ecological study using our model organism, the New Guinean toxic bird *Pachycephala schlegelii* (Regent Whistler) known to integrate the neurotoxic steroid alkaloid batrachotoxin (BTX) into feathers to likely deter predators[2]. Amplicon sequencing allowed us to map the microbial community of UG secretions and feathers of *P. schlegelii* individuals, and guided isolation allowed cultures of microbes with putative protective functions. One isolate belonging to the genus *Amycolatopsis* was singled out for analysis due to its antibacterial activity against keratinolytic bacteria and antifungal activity against skin pathogens. By tracing the bioactivity, we uncovered not only the production of known antibiotics, but also the production of two new compound families of nonribosomal peptide (NRP) origin that had yet unreported structural modifications. By combining phylogenomic, transcriptomic and knock-out studies we identified and verified the biosynthetic gene clusters responsible for the production of the antimicrobial natural products. We also detected molecular ion features of the identified metabolites in extracts of *P. schlegelii* feathers, verifying their presence in the ecological setting where they exert their presumed active role. To our knowledge, this study represents the first example of the isolation of bioactive NRPs from a bird UG isolate, with two compound classes exhibiting unique structural features and biochemistry, and also provides the first evidence that Actinomycetota may play an important role in maintaining healthy plumage, setting the path for further investigations on natural product producers at the bird-microbe interface.

## Results and discussion

### MiSeq microbiome analysis and isolation of bacterial symbionts

Between 2018 and 2019, we collected twelve *P. schlegelii* individuals from three different sampling sites (Fig. 1A, B). All twelve individuals were subjected to UG microbiome studies, and nine out of twelve individuals were also included in analyzes of the feather microbiome (Supplementary Fig. 1, Supplementary Table 1). Illumina MiSeq 16S rRNA amplicon sequencing yielded reads derived from feather samples that were dominated by Pseudomonadota (70.5%), followed by Actinomycetota (10.1%), and Bacteroidetes (5.8%)[3], while the microbiomes of UG were composed primarily of members of Pseudomonadota (74.1%), followed by Firmicutes (15.5%), and Actinomycetota (3.7%) (Fig. 1C, D; Supplementary Data 1), and thus differed notably from temperate bird species[4]. Subsequently, UG secretions of five *P. schlegelii* specimens were investigated for culturable bacterial strains, which yielded 23 morphologically different bacterial colony forming units, of which isolate PS_44_ISF1 was of interest due to its inhibitory

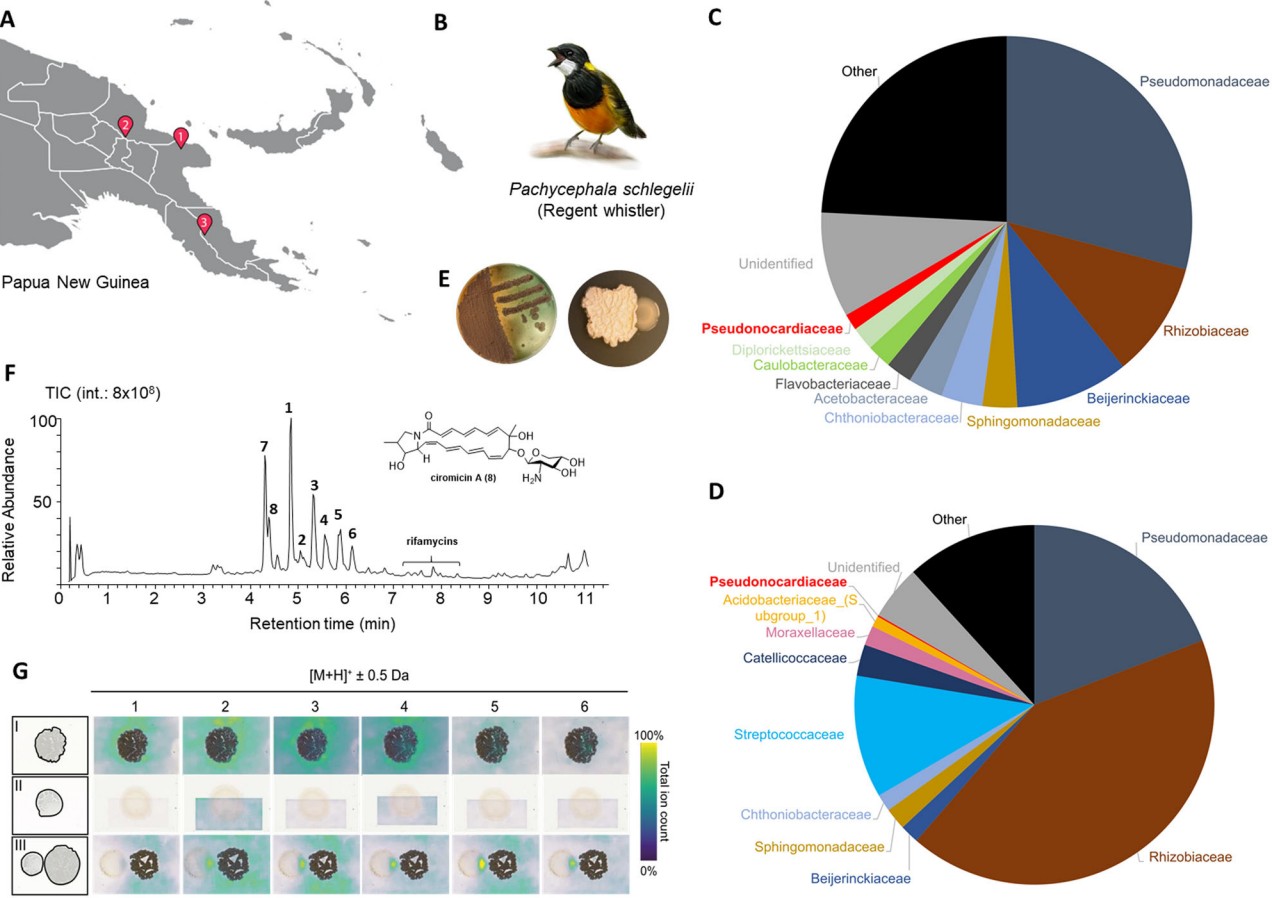

**Fig. 1 | Microbiome study and analysis of the UG isolate *Amycolatopsis* sp. PS_44_ISF1. A** Sampling locations of *P. schlegelii* specimen in Papua New Guinea; **B** Sketch of *P. schlegelii*; and relative averaged abundances of the ten most abundant bacterial families in **C** feather (nine biological samples) and **D** uropygial glands (UG) (twelve biological samples), including in both cases the Pseudonocardiaceae to which *Amycolatopsis* belongs. **E** Representative plate culture of *Amycolatopsis* sp. PS_44_ISF1 alone (top) and in co-cultivation with *B. licheniformis* (bottom). **F** Representative total ion chromatogram (TIC) of methanolic extracts of PS_44_ISF1 showing compounds **1–8** as dominating metabolites with chemical structure of the known metabolite ciromicin A (**8**), and **G** MALDI-IMS images of *Amycolatopsis* sp. PS_44_ISF1 co-cultures with *B. licheniformis* showing the abundance of molecular ion features of compounds **1–6**.

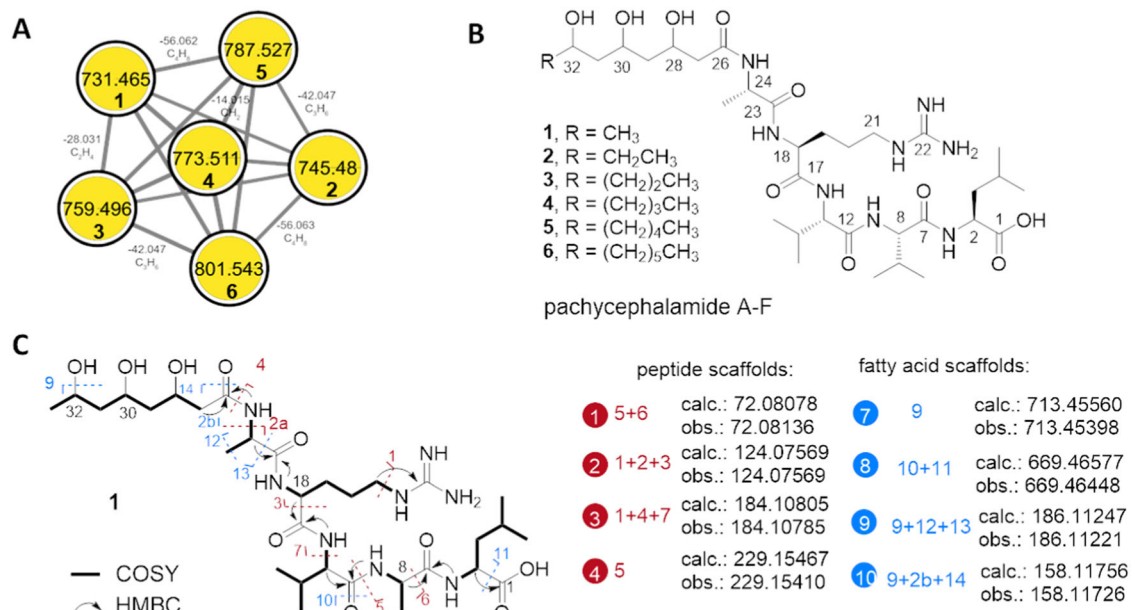

**Fig. 2 | Structure elucidation of pachycephalamides A-F. A** GNPS-based network of molecular ion features of pachycephalamides A-F (**1–6**); **B** chemical structure of **1–6**; **C** 2D-NMR correlations of pachycephalamide A (**1**) and annotated MS/MS fragments with matching calculated *m/z* values predicted by MassFrontier 8.0 (¹H-¹H COSY: Homonuclear correlation spectroscopy; ¹H-¹³C HMBC: Heteronuclear multiple-bond correlation spectroscopy).

activity against other bacterial strains (Fig. 1G). Based on its 16S rRNA sequence, we identified PS_44_ISF1 to be closely related to the type strain *Amycolatopsis saalfeldensis* DSM 44993, which belongs to the family Pseudonocardiaceae (Supplementary Figs. 2 and 3, Supplementary Tables 2 and 3)[16,17].

To enable a more accurate phylogenomic placement of *Amycolatopsis* sp. PS_44_ISF1, a hybrid genome assembly was created by combining paired-end shotgun sequencing with long-read sequencing from PacBio and Nanopore technologies (Supplementary Table 4). Subsequently, whole genome-based taxonomic analysis was performed using the Type (Strain) Genome Server (TYGS)[18] as well as digital DNA-DNA-hybridization (dDDH d4 36.8% to closest strain)[19], which revealed that strain PS_44_ISF1 is likely a new species of the *Amycolatopsis* clade C[20] to which its closest relatives *Amycolatopsis* sp. M39[21] and *A. saalfeldensis* DSM 44993 belong (Supplementary Data 2). This finding was supported by the similar but distinct fatty acid profile of the strain compared to closely related type strains (Supplementary Table 5)[22–24]. To assess whether the isolate could thrive within the UG gland or on feathers, we performed keratinolytic and lipolytic activity tests. In both assessments, *Amycolatopsis* sp. PS_44_ISF1 showed the capability to utilize feather or oil as a carbon source for growth (Supplementary Figs. 4 and 5). We then re-examined if sequencing reads assigned to *Amycolatopsis* were also detectable in our feather and UG microbiome data, and indeed Pseudonocardiaceae accounted for up to 1.5% of the feathers, and 0.14% of the UG microbiome, which included detectable amplicon sequence variants assigned to *Amycolatopsis* in almost half of the samples (Supplementary Fig. 1E). Altogether, our results were intriguing as members of the Pseudonocardiaceae are well recognized for their defensive symbiotic interactions[25], most prominently in the fungus-growing ant symbiosis, where *Pseudonocardia* has aided in protection against specialized parasites of the ants' fungal gardens for millions of years[26,27].

**Antimicrobial properties and secondary metabolome of *Amycolatopsis* sp. PS_44_ISF1**
We then assessed the antimicrobial properties of *Amycolatopsis* sp. PS_44_ISF1 in plate-based co-cultivation assays against ecologically relevant microbial pathogens, including FDB (*B. licheniformis*, *Pseudomonas monteilii*, *K. rhizophila*), non-feather degrading bacteria (*B. thuringiensis*, *Staphylococcus epidermidis*), and fungal pathogens (*Aspergillus fumigatus*, *Aspergillus nidulans*, *Candida albicans*) (Supplementary Figs. 6–20). Indeed, strain PS_44_ISF1 exhibited moderate antifungal activity against the human-pathogenic yeast *C. albicans* and *Aspergillus* spp. as well as growth inhibitory activity against feather-degrading *B. licheniformis* and *K. rhizophila* (Fig. 1G, bottom). We then tested if antimicrobial metabolites caused the observed growth inhibition by analyzing methanolic extracts of the microbial interaction zone using tandem high-resolution mass spectrometry (HR-MS/MS) and global network analyses[28] with interconnected natural-product databases. For dereplication purposes, we also compared the metabolome of *Amycolatopsis* sp. PS_44_ISF1 with two of its close relatives, *Amycolatopsis saalfeldensis*[16,17] and *Amycolatopsis* sp. M39 (Supplementary Fig. 21)[21].

Indeed, a plethora of putative secondary metabolite features were detectable from extracts of *Amycolatopsis* sp. PS_44_ISF1 and the interaction zones of co-cultures, including polyene macrolactam ciromicin A (**8**)[29,30]. Furthermore, database dereplication also uncovered known and PS_44_ISF1-specific molecular ion features belonging to the antibiotic rifamycin family[31,32], but which were in low abundance. However, the number of annotated features was low compared to the abundance of detectable strain-specific unknown features, with two clusters dominating the metabolome. One cluster was comprised of *m/z* features ranging from [M + H]⁺ 731.465 to 801.540 (**1–6**, Fig. 2A) in which nodes differed by Δ*m/z* 14 (CH₂ units), while the second cluster (**7** with [M + H]⁺ 694.351) showed a distinct peptide-related MS/MS pattern (Fig. 3A).

To visualize the molecular ion features in microbial co-culture settings, we also performed Matrix-assisted laser desorption ionization mass spectrometry imaging (MALDI-IMS) analysis of single and co-cultures[33]. Here, we observed in most cases comparable higher abundances of molecular ion features [M + H]⁺ 731.5–801.4 (**1–6**) within the interaction zone of microbial co-cultures in comparison to axenic growth, while the abundance of the molecular ion feature [M + Na]⁺ 716.5 (**7**) and others appeared to remain mainly unchanged (Fig. 1G, Supplementary Figs. 17–20). At this stage, we concluded that

*Amycolatopsis* sp. PS_44_ISF1 produces a series of antimicrobial secondary metabolites that likely account for its observed antimicrobial activity.

## Characterization of peptide-based metabolites

To enable the characterization of the two yet unknown compound families, we evaluated the effect of different growth conditions, including eco-mimetic settings, on their production (Supplementary Figs. 22 and 23). Overall, relative production of both compound families appeared highest on PDA agar plate cultivation between 7 and 10 days. Following up on this finding, large scale cultivation (120 PDA plates, 90 × 16 mm) and methanolic extraction was followed by an MS- and NMR-guided isolation strategy using semi-preparative reversed-phase column (C₁₈) chromatography (HPLC). During the course of purification, we isolated seven yet unreported metabolites that we named pachycephalamides A-F (**1**–**6**, 2.1–7.6 mg) and demiguisin (**7**, 2.7 mg) (Fig. 2).

The molecular formula of pachycephalamide A (**1**) was determined as $C_{33}H_{62}O_{10}N_8$ based on HR-ESI-MS (calcd. [M+H]⁺ *m/z* 731.4662, obsd. [M + H]⁺ *m/z* 731.4653), which indicated seven degrees of unsaturation. The ¹H and ¹³C NMR spectra of **1** (CD₃OH-*d₃*) showed well dispersed proton and carbon resonances indicating a pattern of chemical shifts for a typical peptidic compound as bearing five amide/acid carbonyl signals ($\delta_C$ 175.4, 173.9, 173.8, 173.7, and 173.3), five α-carbon signals ($\delta_C$ 61.2, 61.1, 53.2, 51.7, and 51.3), and five α-proton signals ($\delta_H$ 4.26, 4.16, 4.12, 3.96, and 3.84). In addition to these typical peptidic resonances, pachycephalamide A showed an aliphatic signal pattern including three hydroxyl carbon signals ($\delta_C$ 71.8, 69.8, and 65.9). Further analysis of 1D (¹H and ¹³C) and 2D (COSY, HSQC, and HMBC) NMR data together with MS/MS fragmentation patterns enabled the identification of five amino acid residues (Leu, Ala, Arg, and two Val) and a 3,5,7-trihydroxyoctanoic acid moiety (Supplementary Figs. 24–60, Supplementary Tables 6–8).

As the molecular ion features of pachycephalamide B-F (**2**–**6**) differed by Δ14 Da (CH₂) with otherwise similar fragmentation pattern, it was deduced that the trihydroxyalkyl acid chain varied only by the number of methylene units (Supplementary Methods). The absolute configuration of each amino acid residue in pachycephalamides was analyzed by LC-MS guided Marfey's method using *L*-FDAA (1-fluoro-2,4-dinitrophenyl-5-alanine amide) as the derivatization agent (Supplementary Tables 9 and 10), and determined as *L*-configuration, which is in line with the in silico substrate specificity analysis. Sequence alignment of characterized ketoreductase (KR) domains with the putatively responsible KR domain within the PKS cluster region ctg2_1673 (*vide infra*) inferred an all B-type KR stereospecificity. Thus, we propose that all three hydroxyl moieties exhibit *R*-configuration. Overall, pachycephalamides are lipopentapeptides carrying an unusual odd or even numbered trihydroxylated fatty acid tail, which to the best of our knowledge has not yet been reported.

The molecular formula of demiguisin (**7**) was deduced to be $C_{30}H_{47}N_9O_{10}$ (*m/z* 694.3520 [M + H]⁺, cacld. for $C_{30}H_{48}N_9O_{10}$, 694.3519), indicating 12 degrees of unsaturation (Supplementary Figs. 61–71, Supplementary Table 8). The HSQC NMR spectrum displayed resonances that are typical of a peptide possessing six amino methines, which was consistent with ¹³C NMR signals assigned to six amide/acid carbonyl signals. 1D and 2D NMR data analyses enabled us to identify the presence of three unmodified amino acid residues with a glycine (Gly) moiety at the C-terminus and a leucine (Leu), and piperazic acid (Piz) moiety incorporated into the peptide chain. In addition, atypical amino acid signals such as oxygenated methine correlations and olefin signals were detectable, suggesting the presence of three (oxidatively) modified alkenyl amino acid units. Next to the C-terminal glycine, we deduced the presence of the oxygenated amino acid 4,5-dihydroxyhexahydropyridazine-3-carboxylic acid (β,γ-OH-Piz), for which a characteristic proton spin system consisting of a

methine proton H-4 ($\delta_H$ 4.76), oxygenated methine protons H-5 ($\delta_H$ 4.17) and H-6 ($\delta_H$ 3.39), an methylene proton H₂-7 ($\delta_H$ 3.15 and 2.59), and a secondary amine 7-NH ($\delta_H$ 3.17) (H-4/H-5/H-6/H₂-7/7-NH) was assigned, and which was supported by HMBC correlations from H-4 to C-3 ($\delta_C$ 168.4). The planar structure of the 3,4-dehydropipecolic acid (DHPA, 3,4-dehydro-homoproline) was deduced from COSY and TOCSY correlations between H-20 ($\delta_H$ 5.75)/H-21 ($\delta_H$ 5.88)/H-22 ($\delta_H$ 5.93)/H₂-23 ($\delta_H$ 2.20 and 2.05)/H₂-24 ($\delta_H$ 3.96, 3.38) together with the formation of a spin system ranging from the methine carbon C-20 ($\delta_C$ 51.9) to the epsilon methylene carbon C-24 ($\delta_C$ 40.3). HMBC correlations of the α-proton (H-20, $\delta_H$ 5.75) and δ-proton (H-23b, $\delta_H$ 2.05) to γ-olefin carbon (C-21, $\delta_C$ 122.3), and from ε-proton (H-24a, $\delta_H$ 3.96) to β-olefin carbon (C-22, $\delta_C$ 126.9) aligned with the deductions. A partial structure of the third modified amino acid, γ-carbamoyl-proline (γ-c-Pro), was deduced from COSY and TOCSY correlations as proton–proton couplings between the methine proton H-26 ($\delta_H$ 3.98), the β-protons H₂-27 ($\delta_H$ 2.33 and 1.67), the γ-methine proton H-28 ($\delta_H$ 2.84), and the methylene protons H₂-29 ($\delta_H$ 3.04 and 2.90) (H-26/H₂-27/H-28/H₂-29) were observed. The HMBC correlations from primary amine proton signals (30-NH₂, $\delta_H$ 7.34 and 6.82), β-proton (H-27a, $\delta_H$ 2.33), and δ-proton (H-29a, $\delta_H$ 3.04) to C-30 ($\delta_c$ 175.7) allowed to deduce the planar structure of γ-c-Pro. The amino acid sequence of Gly−β,γ-OH-Piz−Leu−Piz−DHPA−γ-c-Pro (Fig. 3B), which satisfies the 12 degrees of unsaturation, was inferred from the molecular formula, sequence-typical MS/MS fragments and supported by HMBC and ROESY correlations from α-proton signals to amide carbonyl signals of neighboring amino acids. Marfey's analysis allowed again to determine the absolute configuration of the unmodified amino acids Leu and Piz[34] (Supplementary Fig. 61, Supplementary Table 10).

Overall demiguisin (**7**) represents a linear hexapeptide containing three modified amino acid motifs (β,γ-OH-Piz, 3,4-DHPA and γ-carbamoyl-Pro). Peptides containing Pip or Piz, are regularly found to be part of NRPS-derived peptides and known to possess various biological properties, including antibacterial, antifungal and cytotoxic activity, and can act as biofilm inhibitors[34,35].

In contrast, α,β-dehydroamino acids are only occasionally found with α,β-(2,3)-dehydropipecolic acid in ulleungamides as one of the few examples[36–38]. Even less common are examples for peptides containing β,γ-dehydroamino acids or γ-carbamoyl proline. Similarly, γ-hydroxy proline[39] - a major constituent of animal collagen – was found only occasionally as a structural feature in natural products, such as antifungal heinamides[40] and pneumocandins[41], while the structurally close γ-carboxy proline has not yet been reported from NRPS-derived natural products but itself acts as a glutamate uptake inhibitor[42].

## Biosynthetic considerations and validation

To identify the biosynthetic gene clusters (BGCs) responsible for the production of pachycephalamide A-F (**1**–**6**) and demiguisin (**7**), we employed a combination of phylogenomic, transcriptomic and mutagenesis studies (Fig. 4). The genome of isolate PS_44_ISF1 was first mined using the web-based antiSMASH7.0 platform[43] and build-in NRPSpredictor2[44], followed by manual curation of the resulting query hits with respect to the number of modules, adenylation (A) and condensation (C) domains and specificity-conferring codes that matched the amino acid sequence incorporated in the peptide structure (Supplementary Tables 11–13, Supplementary Data 3 and 4). We also performed phylogenetic analyses (maximum likelihood)[45] of encoded and reference domains retrieved from the NaPDos database[46] (Supplementary Figs. 72–75). In parallel, we pursued a transcriptomic analysis of RNAseq data from cultures that were grown on either a lipopeptide production medium (PDA) or low production medium (CSA) and then mapped the transcript levels of the encoded genes within the boundaries of the gene clusters region (Supplementary Figs. 76–78).

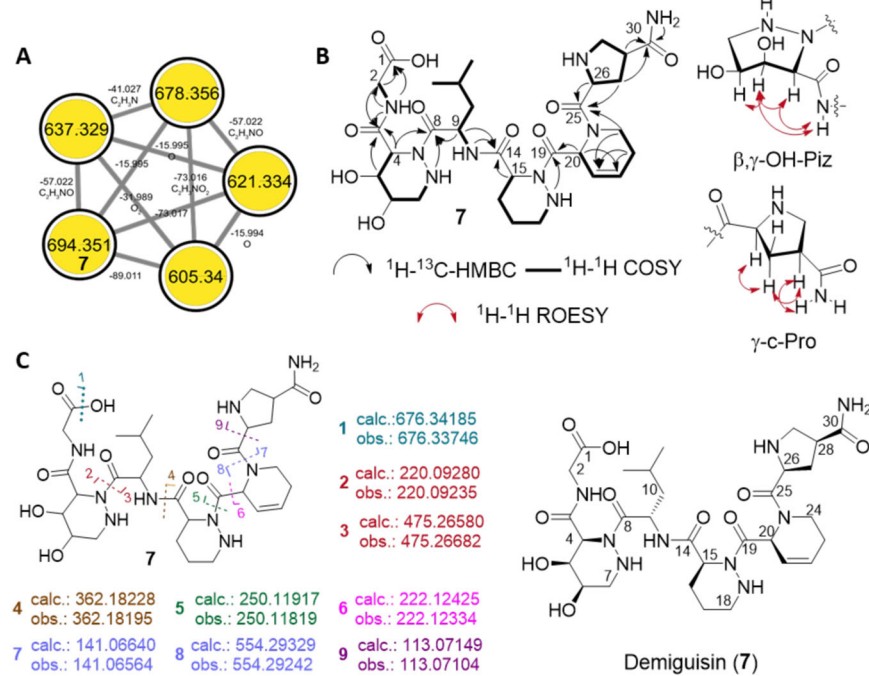

**Fig. 3 | Structure elucidation of peptide demiguisin (7). A** GNPS network showing molecular ion features including the one of demiguisin (7). **B** Proposed absolute structure of **7. C** 2D-NMR correlation of demiguisin and detected MS/MS fragments with atom numbering for NMR assignments. A set of fragments matching calculated *m/z* values predicted by MassFrontier 8.0 (¹H-¹H COSY: homonuclear correlation spectroscopy; ¹H-¹³C HMBC: Heteronuclear multiple-bond correlation spectroscopy; ¹H-¹H ROESY: Rotating-frame nuclear Overhauser effect spectroscopy).

First, we focused on the biosynthesis of the lipopetides pachycephalamides, for which we hypothesized a hybrid gene cluster arrangement composed of a dedicated non-ribosomal peptide synthase (NRPS)-machinery interconnected with either a fatty acid synthase (FAS) or a highly reducing type II PKS enzyme machinery, which was also reported for the structurally related ishigamides and other unsaturated polyene-type natural products[47–51]. Typically, the elongation of the acyl chain through a peptide scaffold necessitates the presence of a starter condensation domain (Cs) located at the initial module of the NRPS assembly line[52,53]. Genome and transcriptome mining uncovered 13 putative NRPS, PKS or NRPS/PKS hybrid candidate cluster regions, but only two of which carried a fitting number of A-domains within an NRPS encoding region (R2.6, R2.11, Supplementary Fig. 77), as well as a neighboring PKS or FAS encoding region. To obtain more insights on the putative candidate clusters, we then analyzed and compared transcript levels of genes within the boundaries of the 13 PKS or NRPS cluster regions (Fig. 4). Two additional NRPS-encoding regions (R2.7 and R5.2) showed differential expression levels and encoded NRPS-like domain architectures that fitted the deduced compound structure. However, none of the regions encoded PKS-related genes up and downstream of the cluster, which led us to hypothesize that these BGC could also be fragmented and/or proceeding via a non-canonical pathway.

We then pursued a knock-out strategy targeting four core NRPS genes and two KS-CLF gene pairs within the four most likely NRPS or PKS-encoding regions, which included the two highly expressed regions R2.7 (*ctg2_971*) and R5.2 (*ctg5_186*), as well as the NRPS/PKS hybrid regions R2.6 (*ctg2_817*) and R2.11 (NRPS: *ctg2_1663*, KS/CLF: *ctg2_1671/2*, *ctg2_1674/5*). For this, a suicide vector was constructed by amplifying an internal fragment (~2 kb) of the target gene, which was then cloned into the SphI/XbaI-linearized *E. coli-Streptomyces* shuttle vector pKJ55 (Fig. 4B). The vector was transferred to *Amycolatopsis* sp. PS_44_ISF1 by conjugation using the methylation-deficient helper strain *E. coli* ET12567/pUZ8002[54,55]. As the vector is not able to

integrate (removed integrase) or replicate, it integrates via heterologous recombination, thereby interrupting the target gene in the *Amycolatopsis* genome (Supplementary Figs. 79–81, Supplementary Tables 14 and 15, Supplementary Data 3). Overall, we were able to obtain knock-outs for all four NRPS-encoding genes and verified apramycin-resistant ex-conjugants were analyzed for the production of pachycephalamides by HR-MS/MS (Supplementary Figs. 82–86). Overall, only one NRPS-knock-out of *ctg2_971* (region 2.7) abolished pachycephalamide production (Fig. 4C), while the relative production levels of other secondary metabolites remained unchanged.

Region 2.7 (*ctg2_971*) carries the BGC cluster now termed *pch*, which spans approx. ~29 kb (20 open reading frames) and encodes eight NRPS-related genes (*pchP1-P8*), four genes putatively annotated as oxidoreductases (*pchOR1-OR4*), four regulator genes (*pchR1-R4*), and several genes with other functions (Fig. 5A). A closer in silico analysis of the encoded NRPS machinery indicated that the four encoded adenylation (A) domains, including one putative iterative domain, showed substrate-conferring codes recognizing Ser (PchP4-A1), Arg (PchP1-A1), Thr (PchP7) and Leu (PchP3-A1). Although PchP4-A1 (Ser) and PchP7 (Thr) showed deviating amino acid conferring codes, A-domain promiscuity[56] might allow the activation of the structurally related Ala and Val, suggesting the formation of the deduced chemical substructure of pachycephalamides. The phylogenetic analyses allowed us to identify a C domain (PchP4-C1) that was classified as an LCL domain (Supplementary Fig. 72), but also showed some relation to the C-domains SpoT10 (sporolide biosynthesis)[57] and SgcC5 (C-1027 biosynthesis)[58], both catalyzing the condensation of two biosynthetic intermediates (Fig. 5C). Thus, we hypothesized that the PchP4 C-domain could catalyze the condensation (lipoinitiation)[53] reaction of the trihydroxylated PKS/ fatty acid precursor and the first amino acid L-Ala, while the second minimal NRPS module encoded by PchP1 could catalyze the downstream condensation with L-Arg (Fig. 5D). The NRPS module PchP8 is assumed to catalyze the subsequent biosynthetic step, despite its unusual architecture (C-PCP-C)

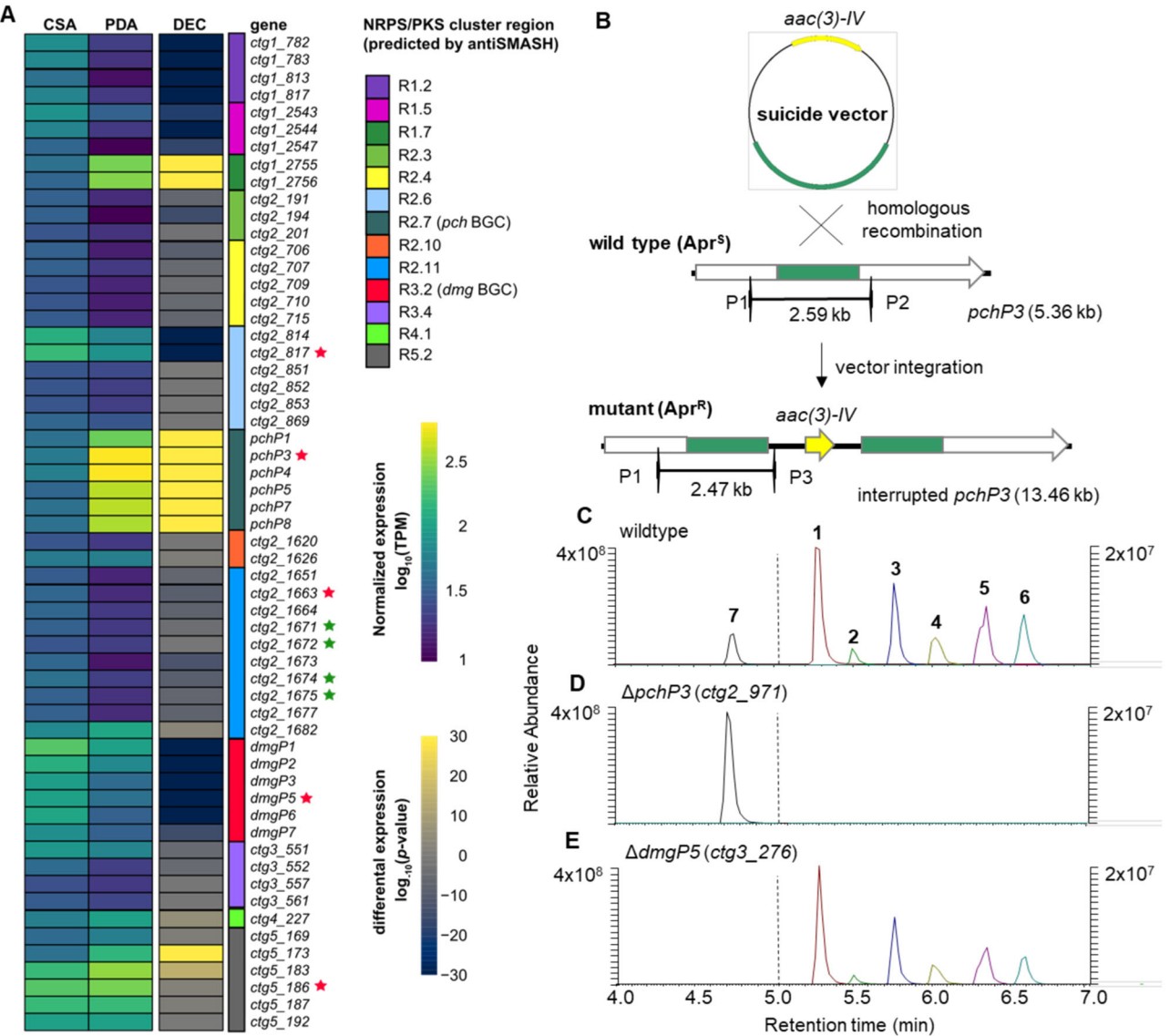

**Fig. 4 | Transcriptomic studies and knock-out experiments for the verification of pachycephalamide and demiguisin biosynthetic gene clusters. A** Analysis of the differential expression of NRPS-encoding cluster regions encoded in the PS_44_ISF1 genome when comparing growth on a lipopeptide high-production (PDA) and low-production (CSA) medium (DEC differential expression confidence, $n = 1$, Supplementary Figs. 76–78, source data are provided as a Source Data file). Five NRPS-encoding genes (red star) and two PKS-CLF gene pairs (green star) were

chosen for further knock-out studies to validate the biosynthetic proposals. **B** Schematic illustration of the generation, selection, and screening of PS_44_ISF1 knock-out mutants (Apr$^S$: apramycin-sensitive, Apr$^R$: apramycin-resistant). **C** Extracted ion chromatograms for the detection of compounds **1**−**7** in the generated knock-out mutants. Compared to the wild-type (**C**), pachycephalamide production was abolished in **D** *ΔpchP3* and production of demiguisin was abolished in mutant **E** *ΔdmgP5*.

lacking adenylation domains. Here, we identified an upstream located stand-alone A domain PchP7 with proposed Thr/Val specificity, which could act iteratively in *trans* and interact with both C domains of PchP7, introducing two Val moieties to the growing peptide scaffold. While iteratively and in *trans* acting domains are unusual, they are not unprecedented. In WS9326A biosynthesis[59], one C-domain was shown to interact successively with two stand-alone A-domains. Lastly, PchP3 is supposed to incorporate Leu, and the mature lipopeptide is likely released by the stand-alone thioesterase PchP5. Although the *pch* cluster encoded a ketosynthase (KS) domain (PchP3) in its proximity, its function in the biosynthesis of the unusual odd and even-numbered 3,5,7-trihydroxylated fatty acids remains hypothetical as it lacked ketoreduction (KR), dehydratation (DH) or enoylreduction (ER) domains. Based on homology searches we propose that the type II PKS of region 2.11 complements the biosynthesis instead, as it harbors two

pairs of type II KS domains accompanied with a chain length factor (CLF) with high similarity to the non-aromatic type II PKS, such as skyllamycin (*sky*), ishigamide (*iga*) and ADEP1 (*ade*) (Fig. 6E)[50,52]. The first CFL/KS pair (*ctg2_1671/2*) co-locates with a discrete KR domain (*ctg2_1673*), while the second pair (*ctg2_1674/5*) co-locates with a DH (*ctg2_1677*) and enoylreductase (ER, *ctg2_1682*) domain. Based on these findings, we hypothesize that the reducing KS-CLF machinery (*ctg2_1674/5*) is able to use acetyl-CoA as well as propionyl-CoA. Two additional elongation steps using malonyl-CoA result ultimately in hexanoyl-ACP or heptenyl-ACP. Assuming that these ACP-bound intermediates are recognized by the partially reducing KS-CLF pair ctg2_1671/2, subsequent Claisen condensations followed by reduction of the carbonyl group would allow introduction of the 3,5,7-trihydroxylation pattern into a variety of different fatty acid intermediates. Based on the alignment of ketoreductases, the KR domain in ctg1_1673

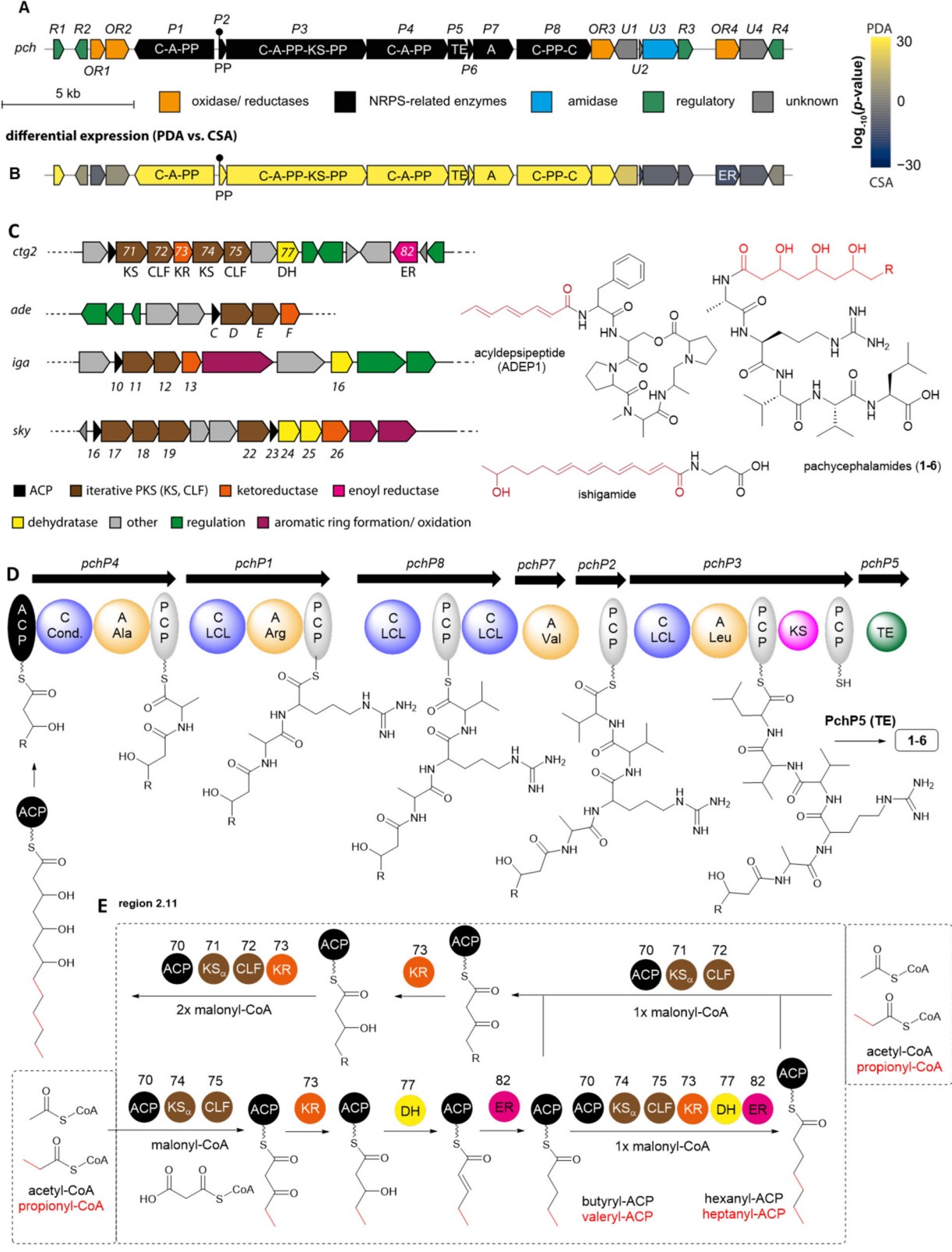

**Fig. 5 | Maps of biosynthetic pathways involved in pachycephalamides bio-synthesis. A** Organization of the *pch* BGC; **B** differential expression of *pch* genes when comparing growth on a lipopeptide high-production (PDA) and low-production (CSA) medium (*n* = 1, source data are provided as a Source Data file); **C** comparison of reducing type II PKS clusters and module numbering with type II PKS clusters of skyllamycin (*sky*), ishigamide (*iga*) and ADEP1 (*ade*) and chemical structures of skyllamycin, ishigamide and ADEP1. **D** NRPS assembly line for the production of pachycephalamides, and **E** interlinked PKS-based pathway respon-sible for the production of 3,5,7-trihydroxy fatty acid with varying $CH_2$-units (ACP acyl carrier protein, KS ketosynthase, CLF chain length factor, DH dehydratase, ER enoyl reductase, TE thioesterase, C condensation domain, A adenylation domain; presentation of stereochemistry was omitted for clarity).

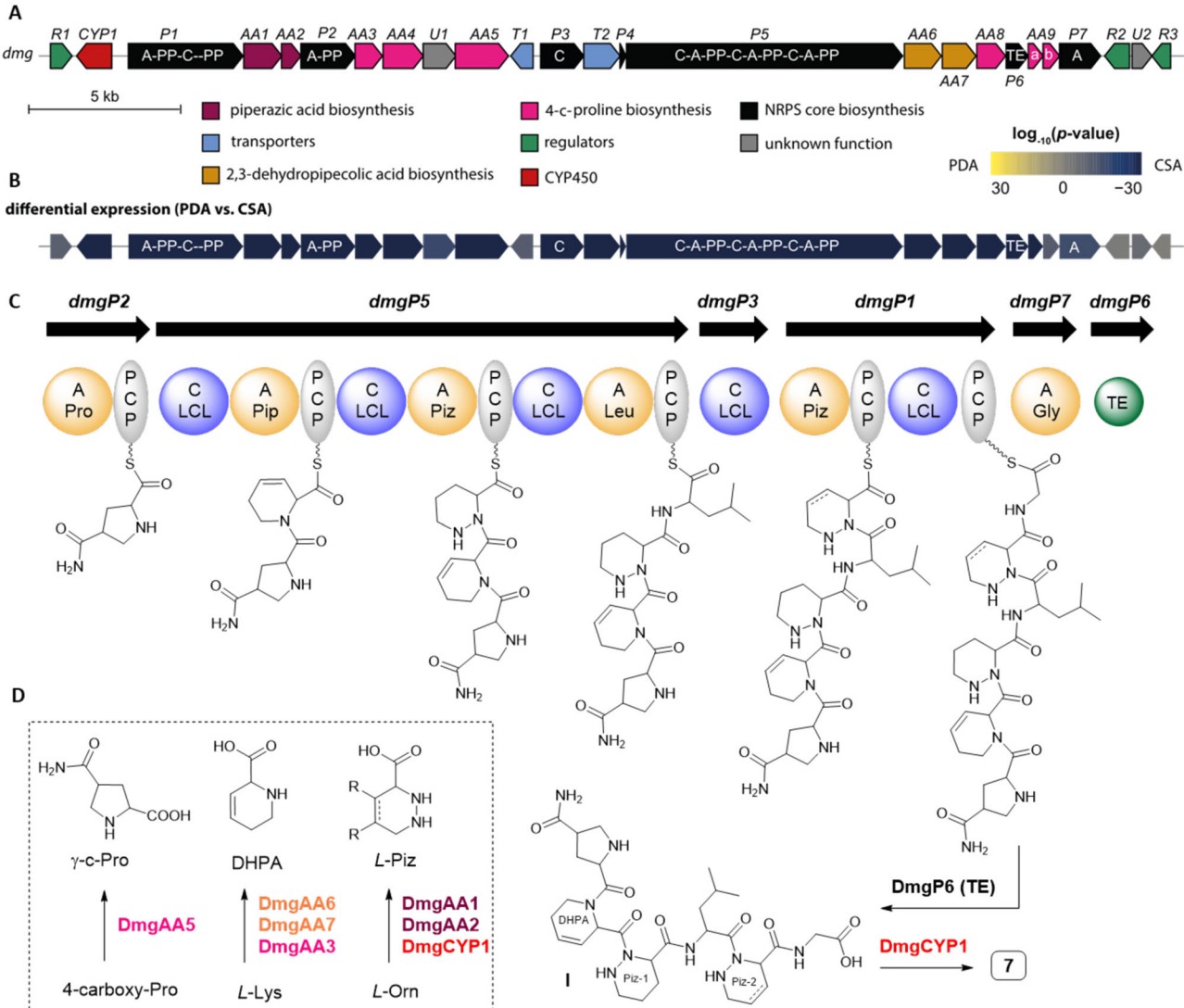

**Fig. 6 | Biosynthesis of demiguisin (7). A** Organization of the *dmg* BGC and **B** differential expression of *dmg* genes when comparing growth on a lipopeptide high-production (PDA) and low-production (CSA) medium (Source data are provided as a Source Data file, *n* = 1); **C** NRPS assembly line for the production of a proposed demiguisin precursor **I**, including the origin of the modified amino acid precursors (Supplementary Fig. 87), and **D** proposed transformation/enzymes for precursor biosynthesis (PCP peptidyl carrier protein, C condensation domain, A adenylation domain, TE thioesterase, Pip piperazine, Piz piperazic acid, Gly glycine, Pro proline, Lys lysine, Leu leucine, Orn ornithine, presentation of stereochemistry was omitted for clarity).

was determined as B-type, thus the stereochemistry of three hydroxyl moieties is proposed to be *R* configured (Supplementary Fig. 75)[60]. In a final step, the respective acyl chains could be linked to the peptide backbone by the first C domain in the pachycephalamide NRPS assembly line.

For demiguisin (**7**) we identified one gene cluster region (*dmg*) encoding a multimodular NRPS carrying six A domains, which we deemed responsible for its biosynthesis (Supplementary Data 4). The gene cluster region *dmg* spanned about 38.2 kb and encoded at least 25 open reading frames (*dmgR1-R3*), including seven NRPS-related genes (*dmgP1-P7*), nine genes encoding enzymes responsible for building block formation and modification (*dmgAA1-AA9, dmgCYP1*), five genes coding for transport (*dmgT1-T2*) and regulatory (*dmgR1-R3*) functions, among others (Fig. 6). Transcripts of all genes located within the boundaries of the BGC region were found in RNAseq data from two growth conditions, which correlated with production of demiguisin (**7**). We then analyzed the A domains encoded in the cluster region for their substrate specificities and found that four of the stand-alone domains showed specificity for Pro/Pip (DmgP1-A1, DmgP2-A1, and

DmgP5-A1/A2), one for Leu (DmgP5-A3) and one for Gly (DmgP7-A1). Of the four A domains proposed to recognize Pro or Pip, two shared a 90% identical specificity-conferring code and were also related to the consensus code of Piz-activating A domains (DmgP1-A1 and DmgP5-A2, Supplementary Table 13)[34,35].

To verify that this particular NRPS is responsible for the assembly of the peptide backbone of demiguisin (**7**), we again prepared knockout mutants by interrupting the key NRPS-encoding gene within the *dmg* cluster (*dmgP5*). As expected, demiguisin production in apramycin-resistant ex-conjugants carrying the interrupted NRPS-encoding gene *dmgP5* was completely abolished (Supplementary Figs. 82–86), while the relative production levels of other secondary metabolites remained unchanged, thus supporting the theory that the *dmg* cluster should be responsible for the biosynthesis of the peptide scaffold of **7**.

Based on this dataset, the biosynthetic assembly is proposed as follows (Fig. 6C). The core structure of **7** is likely build first, by activation and loading of the γ-carbamoyl-Pro or γ-carboxyl-Pro (γ-c-Pro) on to NRPS DmgP2, followed by transfer of the PCP tethered γ-c-Pro to

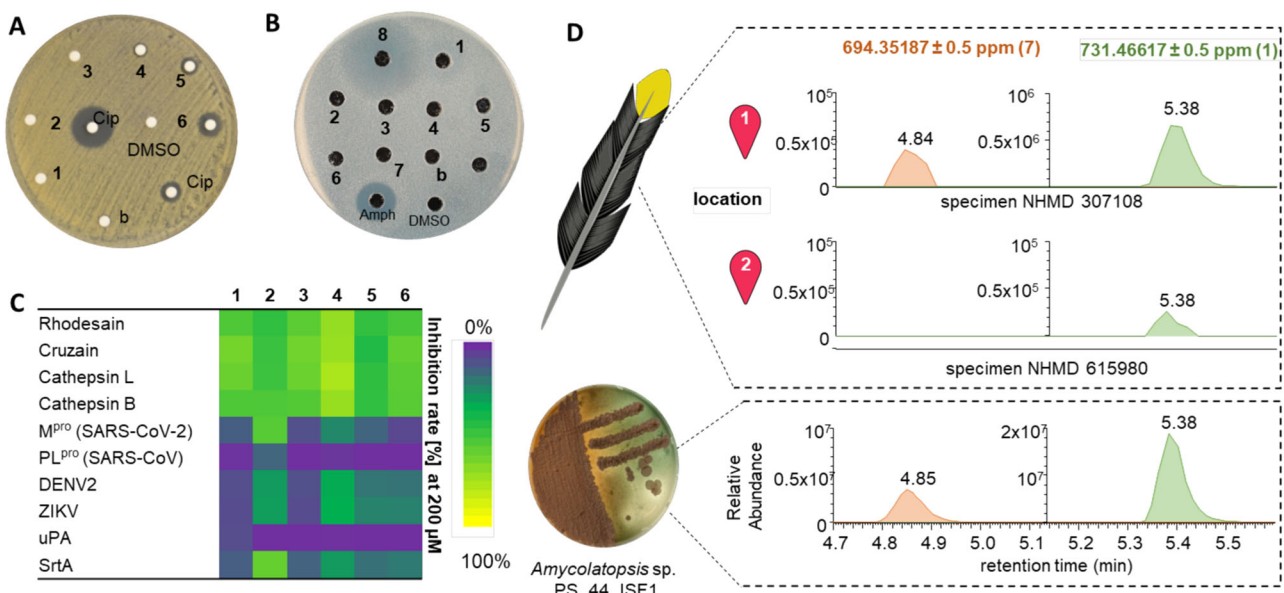

**Fig. 7 | Bioactivity and ecological relevance of compounds.** Disc-diffusion assays of isolated compounds **1–8** against **A** *K. rhizophila* and **B** *C. albicans* (DMSO, blanks (b), and antibiotic controls (Ciprofloxacin: AmpH: Amphotericin B). **C** Inhibition of selected proteases [%] by compounds **1–6** (200 μM, n = 3). **D** Detection of compounds **1** and **7** in methanolic extracts of feathers (sketch) of two *P. schlegelii* specimens from different locations with a culture extract of strain PS_44_ISF1 as reference.

DmgP5 and the step-wise incorporation of the amino acids DHPA, Piz and Leu (Fig. 6). The second (modified) Piz moiety is presumably activated by the A domain of DmgP1 and coupled to the growing peptide backbone via the catalytic action of the stand-alone C domain DmgP3. Glycine is likely activated by the discrete A domain DmgP7 and incorporated by the C domain of DmgP1. The discrete thioesterase DmgP6 should release the mature peptide backbone. All of the necessary C domains of the NRPS modules were found to cluster with known LCL domains, which supported our deduction from prior analytical datasets (Supplementary Fig. 72). To which degree the respective (discrete) A domains are capable of incorporating natural and modified amino acids or whether the peptide chain is modified after the release from the NRPS remains a topic of investigations.

We also identified genes encoding for the necessary precursor biosynthesis within the boundaries of the cluster (Fig. 6D, Supplementary Fig. 87, Supplementary Data 4). Piperazic acid (Piz) biosynthesis is likely catalyzed by DmgAA1-AA2, which show similarities to ʟ-ornithine-N-hydroxylase KtzI (51%) and the heme-dependent piperazate synthase KtzT (55%), commonly present in BGCs of Piz-containing natural products[34,35]. Here, we propose that β,γ-hydroxylation of the Piz-moiety might occur via the action of cytochrome P450 monooxygenase DmgCYP1 either before incorporation or after the release from the NRPS. 3,4-Dehydropipecolic acid (DHPA) is likely formed from ʟ-lysine via α-transamination of α-ketoglutarate catalyzed by the putative ʟ-lysine-2-aminotransferase DmgAA6, which shows similarities to the aminotransferases NikC or CrmC reported in the nikkomycin[61] or caerulomycin A[62] biosynthesis (Supplementary Data 4). As proposed in the nikkomycin biosynthesis, the α-keto intermediate spontaneously cyclizes and dehydrates to Δ¹- or Δ²-piperideine-2-carboxylate (P2C), which can be further oxidized by FAD-dependent oxidoreductase DmgAA7 homologous to NikD/CrmD resulting in a Δ¹-Δ³-P2C intermediate. The encoded reductase DmgAA3 is proposed to reduce the Δ¹ double bond, yielding DHPA.

The biosynthesis of the γ-carbamoyl proline (γ-c-Pro) moiety remains unclear at this stage as two possible biosynthetic scenarios were deduced from our in silco analysis. In the first possible scenario, condensation reaction of ʟ-Ser with oxalacetate by DmgAA4, an enzyme homologous to tryptophan synthase-like enzyme CourR1,

should yield 4-amino-1-oxobutane-1,2,4-tricarboxylic acid, which is then decarboxylated via the action of the decarboxylase DmgAA9a/b to yield Δ⁵-4-carboxyproline (Supplementary Fig. 74). Reduction of the Δ⁵-double bond could be induced by DmgAA3, an oxidoreductase sharing homology with GriH in griselimycin or HrmD in hormaomycin biosynthesis[63]. In an alternative biosynthetic route Leu is first hydroxylated by hydroxylase GriE and then subsequently oxidized to an aldehyde by dehydrogenase GriF. The resulting intermediate cyclizes by forming Δ⁵-4-methyl-Pro which is reduced to 4-methyl-Pro by GriH and is then oxidized to 4-carboxy-Pro via 4-hydroxymethylproline. In both scenarios, amidation of 4-carboxy-Pro could occur either before or after release from the NRPS via the putative amidotransferase DmgAA5, a homolog of amidotransferase SsfD reported in the SF2575 biosynthesis[64].

## Antimicrobial activity and occurrence in feathers

Peptides containing Pip, Piz or α-3,4-DHPA, similar to lipopeptides, are well known to have antimicrobial activities and often confer functions in cell motility and biofilm formation[65]. We thus subjected the isolated compounds (**1–7**) to various different bioassays to explore their activity range and pharmacological properties.

For lipopeptides (**1–6**), we found moderate antibacterial activity against several indicator strains as well as the feather-degrading species *K. rhizophila* (Fig. 7A, Supplementary Table 16), which are likely caused by the amphiphilic character and the chain-length dependent tensioactive properties of this substance family (Fig. 7A, Supplementary Fig. 88)[65,66]. In contrast, demiguisin (**7**) appeared to have no antimicrobial effects, while the co-eluting macrolactam ciromicin A (**8**) and rifamycin congeners exhibited antifungal and antibacterial effects as previously reported (Fig. 7B, Supplementary Fig. 89)[30].

Due to the antibacterial properties of pachycephalamide, we went on to test **1–6** against a panel of different proteases of pharmacological and ecological relevance (Fig. 7C, Supplementary Tables 17–19), including the bacterial transpeptidase sortase A of *Staphylococcus aureus*[67], which is a well-known virulence factor during infections with these bacteria. In addition, we included cathepsin L, cathepsin B[68] and urokinase (uPA) into the screen[69], as they are related to many cancer types and tumor-associated fibroblasts and macrophages. We also

included cathepsin-like proteases rhodesain[70] and cruzain[71], which are central in the life cycle of the parasitic hemoflagellate protists *Trypanosoma brucei* and *Trypanosoma cruzi*, the major cause of African trypanosomiasis (*T. brucei*), also known as "sleeping sickness", and Chagas disease (American trypanosomiasis, *T. cruzi*). Validated targets for new antivirals, such as the proteases (M[pro], PL[pro]) of SARS-coronavirus (SARS-CoV) and SARS-CoV-2[72] and the NS2B-NS3 proteases of the flaviviruses ZIKV and DENV[73] (Zika and Dengue) were also tested. A single-concentration assay (200 μM) uncovered that compounds **1–6** moderately inhibited the structurally related cysteine proteases rhodesain, cruzain, cathespin L and cathepsin B in a structure-related fashion. Finally, we tested if metabolic features of *Amycolatopsis* sp. PS_44_ISF1 were also detectable in methanolic extracts of feather samples of *P. schlegelii* specimens[2]. Indeed, targeted analysis revealed molecular ion features of the most abundant secondary metabolites **1** and **7** in two specimens collected from two different regions with the same retention time and fragmentation pattern as the isolated and characterized compounds (Fig. 7D, Supplementary Fig. 71).

Wild bird species host intricate microbial communities throughout various body regions, yet our comprehension of the roles these microbiomes play trails behind that of many other vertebrates. Following up on our findings that the relative microbial composition between the UG system and feathers of the toxic bird species *P. schlegelii* show notable differences, we were able to culture a fraction of the predicted UG and feather microbiome including a new member and likely a new species of the genus *Amycolatopsis,* which includes species well recognized for their defensive symbiotic interactions and protection against specialized parasites. With *Amycolatopsis* sp. PS_44_ISF1 as a representative of the bird protective microbiome, we analyzed its keratinolytic and lipolytic activities and found evidences for its ability to thrive in the bird gland and feather environment. Subsequent co-cultivation experiments coupled with MALDI-IMS analysis revealed the strains ability to suppress common bacterial and fungal pathogens found in birds, including FDB. The observed antimicrobial activity was attributed to the co-secretion of several classes of natural products, including the antimicrobial polyene macrolactam ciromicin A, the low-abundant ansamycin-derived rifamycin congeners, as well as the two newly discovered peptide classes of NRPS origin, both exhibiting unique structural features. While pachycephalamides A-F (**1–6**) represent pentapeptides with a 3,5,7-trihydroxylated fatty acid ($C_8$-$C_{13}$) moiety linked to the N-terminus, demiguisin (**7**) is a hexapeptide containing various modified amino acids, such as β, γ-hydroxylated piperazic acid, dehydropipecolic acid γ-carbamoylproline. To the best of our knowledge, these represent the first examples of NRPS-derived natural products isolated from a bacterial strain originating from an UG. We further were able to detect members of both peptide classes in feathers of *P. schlegelii,* supporting that the strain and its metabolome might play a beneficial protective role for the host bird. Altogether, our results indicate that along with gland secretions, bacteria derived from uropygial glands and secreting antimicrobial natural products are likely applied onto the feathers to inhibit the growth of harmful microbes.

By integrating phylogenetic, transcriptomic, and mutagenetic studies, we propose that two predominantly non-canonical biosynthetic gene clusters are accountable for the biosynthesis of the two new compound classes. Both clusters include *trans*-acting stand-alone as well as iteratively acting NRPS domains, and a fascinating complementary pair of (iterative) PKS modules. Our molecular biological studies serve now as the foundation for further elucidation of the enzymatic reactions responsible for producing these unusual features. A deeper understanding of these mechanisms will facilitate targeted efforts to identify and exploit these unique features in other natural product producers. Overall, our study also reaffirms the potential inherently present when exploring microbial communities in

underexplored ecological niches, offering opportunities for the identification of new bacterial species, innovative natural product producers, and previously undisclosed biochemical processes.

## Methods

### Ethics

The fieldwork protocol, including ethical standards follows Guidelines to the Use of Wild Birds in Research formulated by The Ornithological Council. Samples were collected in 2018 and 2019 at three locations in Papua New Guinea (PNG) under the research permits no. 99902341112 (to Kasun Bodawatta) and no. 99902260244 (to Knud Jønsson) and exported under export permits no. 019067 in 2018 and no. 019362 and no. 019423 in 2019 by the PNG Immigration & Citizen Service Authority and the Department of Environment and Conservation, Boroko, Papua New Guinea.

### MiSeq sequencing of UG and feather samples

UGs were collected from thirteen *P. schlegelii* specimens obtained from three locations in Papua New Guinea under the research permit 99902341112 in the years 2018 and 2019. Birds were humanely euthanized through rapid cardiac compression following the Natural History Museum of Denmark collection guidelines, and directly dissected on a sterile field tray. While UG were stored in RNALater™ until DNA extractions, ~20 μg of breast feather samples from each of the individual were stored in individual Ziploc bags, initially, at room temperature (approx. 20 °C), and once reached the field station, samples were kept in −20 °C until further analyses. DNA from UG and a subset of the feather samples (4–5 breast feathers) were extracted following the manufacture protocol. Samples were sent to the Microbiome Core at the University of Michigan for MiSeq amplicon sequencing, using primer pair SA701 and SB511 targeting the V4 region of the bacterial 16s rRNA gene under an Illumina sequencing platform. For more details, see Supplementary Methods.

### Analysis of MiSeq sequencing data

Miseq amplicon sequences were analyzed using the QIIME2[74] and DADA2[75] pipelines with default settings. Sequences were assigned to amplicon sequence variants (ASVs) at the 100% similarity and subsequently assigned to taxonomy using the SILVA 138.1 bacterial reference database[76]. We further removed archaeal, mitochondrial and chloroplast sequences from the dataset. Using the phyloseq package[77] we analyzed the bacterial alpha diversities (chao 1 richness estimate and Shannon's diversity index) of the microbiomes. We visualized the microbial community compositions (based on Bray-Curtis dissimilarity) using NMDS plots and investigated the statistical difference between microbiomes using adonis 2 test (Permutational multivariate analyses of variance: PERMANOVA) in vegan package (https://CRAN.R-project.org/package=vegan).

### Isolation of UG-associated bacteria

UG secretions were applied directly onto a Potato Dextrose Agar medium (PDA, 32 g of PDA mixed with 800 mL of water) containing 50 mg cycloheximide per liter of media. Plates were kept in room temperature under aerobic conditions. Once the colony forming units (CFUs) appeared, morphologically different colonies were isolated and reinoculated in new PDA media without cycloheximide.

### Strain identification by amplicon sequencing

Strain PS_44_ISF1 was sub-cultured on MS agar, and single colonies stored in 25% glycerol at −80 °C. Bacterial DNA from the strain was acquired using Qiagen DNeasy® Blood and Tissue kit (Hilden, Germany), with a 24-h incubation period. Initial Sanger sequencing was conducted at the Eurofins genomics (Copenhagen, Denmark), with the bacterial 16S rRNA gene primer pair 27 F and 1492 R to identify the bacterial isolate[78].

## Whole genome sequencing and analysis

*Amycolatopsis* sp. PS_44_ISF1 cultures (50 mL) were prepared in ISP2 broth and cultivated for seven days at 30 °C and shaking at 150 rpm. Bacterial pellets were harvested, frozen in liquid nitrogen and grounded to a fine powder. DNA was extracted using DNeasy Plant Mini Kit (Quiagen) according to manufacturer's instructions and subsequently precipitated using isopropanol. The precipitated DNA was washed with 95% ethanol (at least 5 times), dried and dissolved in water before sequencing. Whole-genome sequencing was performed using a 150 bp paired-end shotgun (BGIseq) and long-read (PacBio sequel) sequencing at BGI. Additionally, Oxford Nanopore technology (Oxford Nanopore Technologies, Oxford, UK) was employed for long-read sequencing. The MinION sequencing library was prepared using the Rapid DNA sequencing kit (SQK-RAD4) according to the manufacturer. DNA sequencing was performed on a MinION Mk1B sequencing device equipped with a R9.4.1 flow cell, which was prepared and run according to the manufacturer.

Sequencing results were checked for quality using FastQC version 0.11.8[79] and MultiQC version 1.7[80]. K-mer depth was calculated using Jellyfish version 2.2.10[81] and Kmer-based estimates of genome size, heterozygosity, and repeat content generated using GenomeScope[82]. Nanopore sequencing raw data was generated using MinKNOW software version 4.0.20 (Oxford Nanopore Technologies), and was base-called and trimmed using Guppy version 4.2.2 (Oxford Nanopore Technologies). The resulting fastq files were filtered using Nanofilt[83]. A hybrid de novo genome assembly, combining PacBio and Oxford Nanopore data, was performed using MaSuRCA version 3.4.1[84]. The resulting draft assembly was then polished with the accurate Illumina reads using the POLCA genome polisher[85].

## HR-UPLC-ESI-MS/MS analysis

High resolution ultra-high performance liquid chromatography electrospray ionization mass spectrometry (HR-UPLC-ESI-MS) measurements were carried out on a Dionex Ultimate3000 system (Thermo Scientific) combined with a Q-Exactive Plus mass spectrometer (Thermo Scientific) equipped with an electrospray ion (ESI) source and a Luna Omega C18 column (100 ×2.1 mm, particle size 1.6 µm, pore diameter 100 Å). For structure elucidation MS/MS spectra were predicted using CFM-ID 4.0 and compared to the experimental MS/MS spectra. Each sample was analyzed using HR-UPLC-MS$^2$ measurements using a Dionex Ultimate3000 system (Thermo Scientific) (gradient: 0–1 min, 5% B, 1–7 min, 5–97% B, 7–9 min, 97% B, 9–10 min; 97–5% B, 10–13 min, 5% B) with an injection volume of 5 µl and a flow rate of 0.3 mL/min. All eluents of the UHPLC system were acidified with 0.1% formic acid. $H_2O$ was used as eluent A while acetonitrile was used as eluent B. For each analytical runs, washing steps and blank runs were included (at least every 5 biological samples) and served as control samples to check for cross-contamination. For biological samples one technical replicate was measured. For structure elucidation MS/MS spectra were predicted using CFM-ID 4.0 or MassFrontier 8.0 (Thermo Fisher Scientific and compared to the experimental MS/MS spectra.

## Network-based analysis

Tandem mass spectrometry molecular networks were created using the GNPS platform (http://gnps.ucsd.edu)[28]. Data (.RAW) were first converted to the .mzML format with MS-Convert[86]. The converted files were used to generate an MS/MS molecular network using the GNPS Data Analysis workflow version release 30. The precursor and fragment ion mass tolerance were set to 0.02 Da and to a product ion tolerance of 0.02 Da. Networks were generated using six minimum matched ion fragments, a minimum cluster size of 2 and a cosine score of 0.8. The remaining parameters were kept at default. The library spectra were filtered in the same manner as the input data. All matches kept between network spectra and library spectra were required to have a score above 0.7 and at least 6 matched peaks.

After analysis, data were opened and visualized using Cytoscape 3.8.0 software.

## Co-cultivation experiments

*Amycolatopsis* sp. PS_44_ISF1 was cultivated on MS agar for 7 days at 30 °C. Indium-Tin-Oxide (ITO)-coated slides (IntelliSlides, Bruker) were coated by aluminum foil and sterilized by autoclaving. Slides were placed into a petri dish (90 × 16 mm) and 10 ml of PDA were distributed evenly on top of the slide. A spore solution (5 µL, OD$_{600}$ of 0.1) was placed in the center of the PDA-coated slides. *Amycolatopsis* cultures were incubated at 30 °C for a total period of six days. After one (=day 2) and three days (=day 4) of incubation 5 µl of a competitor strain culture or spore solution (OD$_{600}$ = 0.1) was added in a proximity of 1 cm to the *Amycolatopsis* culture. Competitor strains used included three feather-degrading bacteria (*B. licheniformis* DSM13, *K. rhizophila* DSM11926, *P. monteilii* DSM1388), filamentous fungi (*A. fumigatus*, *A. niger*) and the yeast *C. albicans* SC5314. Incubation of co-cultures were pursued until the end of the experiment. Axenic cultures of *Amycolatopsis* sp. PS_44_ISF1 and each competitor strain inoculated at the same time as the co-cultures served as controls. For analysis of metabolite abundance, a defined agar piece was extracted in a pre-weighed beaker after a given time of incubation using MeOH overnight at room temperature. The resulting methanolic extract was dried in *vacuo* and the resulting residue was dissolved in 100% MeOH to a final concentration of 50 µg/ml and subjected to HR-UPLC-ESI-MS analysis. The production of each metabolite of interest was evaluated by the comparison of the peak area using the extracted ion mode (EIC) and normalized to the wet weight of agar extracted (peak area/mg wet weight).

## Matrix-assisted laser desorption ionization mass spectrometry imaging (MALDI-IMS)

Cultures on Indium-Tin-Oxide (ITO)-coated slides were dried for 5 h at 37 °C. The dried samples were then sprayed with a saturated solution (20 mg/mL) of universal MALDI matrix (1:1 mixture of 2,5-dihydroxybenzoic acid and α-cyano-4-hydroxy-cinnamic acid; Bruker Daltonics, Bremen Germany) prepared in acetonitrile/methanol/water (70:25:5, v/v/v), using the automatic system ImagePrep device 2.0 (Bruker Daltonics, Bremen Germany) in 60 consecutive cycles (the sample was rotated 180° after 30 cycles) of 41 s (1 s spraying, 10 s incubation time, and 30 s of active drying)[87]. The sample was analyzed in an UltrafleXtreme MALDI TOF/TOF (Bruker Daltonics, Bremen Germany). For further details, see Supplementary Methods.

## Monitoring of product formation under different growth conditions

For analysis of production levels, *Amycolatopsis* sp. PS_44_ISF1 pre-cultures (50 ml) were prepared in ISP2 broth and cultivated for seven days at 30 °C and shaking at 150 rpm. Agar plates of different media were inoculated with 100 µl of the pre-culture and further incubated at 30 °C for a given time period. Three replicates of each medium were extracted after a given time of incubation by cutting agar into small pieces (0.5 cm × 0.5 cm). After transfer into a pre-weighed 100 mL beaker, the weight of the transferred agar was determined. Agar pieces were extracted with 40 mL MeOH overnight at room temperature. The solvent was then filtered off and evaporated *in vacuo*. The resulting residue was dissolved in 100% MeOH to a final concentration of 50 µg/ml and subjected to HR-UPLC-ESI-MS analysis. The production of each metabolite of interest was evaluated by the comparison of the peak area using the extracted ion mode (EIC) and normalized to the wet weight of agar extracted (peak area/mg wet weight).

## Large scale cultivation for natural product isolation

For isolation from agar plates, 500 µl 7-day-old pre-cultures of *Amycolatopsis* spp. were streaked on 120 PDA (peptides) agar plates (90 × 16 mm). The cultures were sealed with parafilm and incubated at

30 °C for another 7 days. Plates were cut in small cubes (1 × 1 cm) and extracted with 6 L of methanol overnight. Afterwards, the solvent was filtered off and evaporated *in vacuo*. The dried extract was then redissolved in 20% methanol (MeOH) and subjected to solid phase extraction (SPE). A 10 g C$_{18}$ column was activated with 2 column volumes (CV) of 100% MeOH and equilibrated with 20% MeOH in dH$_2$O. Afterwards the extract was loaded, washed with 20% MeOH, and eluted with 50% and 100% MeOH (2 CV). All three fractions were concentrated *in vacuo* yielding all together 98.5 g of extract. For peptide isolation, the combined 50% and 100% fractions were used for further processing and first resuspended with Celite in MeOH and then dried *in vacuo* to generate Celite-adsorbed extract in a round flask. The Celite-adsorbed extract was loaded onto 20 g of the pre-packed C18 SPE Sepak resin. The extract was fractionated by elution with a step gradient composed of dH$_2$O and methanol (10%, 20%, 40%, 60%, 80%, and 100% MeOH). Macrolactam ciromicin A and demiguisin (**7**) eluted in the 40% MeOH SPE fractions, while pachycephalamides A-F (**1**–**6**) eluted in the 60% and 80% MeOH SPE fractions, and rifamycin congeners in the 100% MeOH SPE fractions. Fractions were purified by semipreparative reversed-phase HPLC (Phenomenex Luna C18(2) 250 ×10 mm column, particle size 5 μm, pore diameter 100 Å, flow rate: 2 mL/min, detection: UV 210 nm, gradient solvent system: 10–40% aqueous acetonitrile over 30 min). For further analytical experimental details, see Supplementary Methods.

Pachycephalamide A (**1**): white solid; $[\alpha]_D^{25}$ −5.67 (c 0.1 w/v%, MeOH); UV (acetonitrile), λ$_{max}$ 204 nm; IR (ATR) ν$_{max}$ 3778, 3213, 2783, 2363, 1694, 1550, 1021 cm$^{-1}$; NMR spectral data, see Supplementary Tables 6 and 7; HRMS (ESI) *m/z* [M + H]$^+$ calcd. for C$_{33}$H$_{63}$O$_{10}$N$_8$ 731.4662, found 731.4653, Δ$_{ppm}$ −0.84.

Pachycephalamide B (**2**): white solid; $[\alpha]_D^{25}$ −2.28 (c 0.1 w/v%, MeOH); UV (acetonitrile), λ$_{max}$ 204 nm; IR (ATR) ν$_{max}$ 3743, 3621, 3311, 2702, 2322, 1616, 1562 cm$^{-1}$; NMR spectral data, see Supplementary Tables 6 and 7; HRMS (ESI) *m/z* [M + H]$^+$ calcd. for C$_{34}$H$_{65}$O$_{10}$N$_8$ 745.4829, found 745.4802, Δ$_{ppm}$ −2.81.

Pachycephalamide C (**3**): white solid; $[\alpha]_D^{25}$ −135.89 (c 0.1 w/v%, MeOH); UV (acetonitrile), λ$_{max}$ 204 nm; IR (ATR) ν$_{max}$ 3742, 2940, 2540, 2362, 2162, 1725, 1549, 1022 cm$^{-1}$; NMR spectral data see Supplementary Tables 6 and 7; HRMS (ESI) *m/z* [M + H]$^+$ calcd. for C$_{35}$H$_{67}$O$_{10}$N$_8$ 759.4985, found 759.4968, Δ$_{ppm}$ −2.23.

Pachycephalamide D (**4**): white solid; $[\alpha]_D^{25}$ −145.68 (c 0.1 w/v%, MeOH); UV (acetonitrile), λ$_{max}$ 204 nm; IR (ATR) ν$_{max}$ 3742, 2942, 2540, 2362, 2162, 2024, 1693, 1549, 1021 cm$^{-1}$; NMR spectral data, see Supplementary Tables 6 and 7; HRMS (ESI) *m/z* [M + H]$^+$ calcd. for C$_{36}$H$_{69}$O$_{10}$N$_8$ 773.5131, found 773.5112, Δ$_{ppm}$ −1.88.

Pachycephalamide E (**5**): white solid; $[\alpha]_D^{25}$ −129.60 (c 0.1 w/v%, MeOH); UV (acetonitrile), λ$_{max}$ 204 nm; IR (ATR) ν$_{max}$ 3743, 3621, 3311, 2702, 2322, 1616, 1562 cm$^{-1}$; NMR spectral data, see Supplementary Tables 6 and 7; HRMS (ESI) *m/z* [M + H]$^+$ calcd. for C$_{37}$H$_{71}$O$_{10}$N$_8$ 787.5298, found 787.5278, Δ$_{ppm}$ −2.53.

Pachycephalamide F (**6**): white solid; $[\alpha]_D^{25}$ −196.30 (c 0.1 w/v%, MeOH); UV (acetonitrile), λ$_{max}$ 204 nm; IR (ATR) ν$_{max}$ 3741, 2939, 2540, 2274, 1693, 1549, 1448, 1022 cm$^{-1}$; NMR spectral data, see Supplementary Tables 6 and 7; HRMS (ESI) *m/z* [M + H]$^+$ calcd. for C$_{38}$H$_{73}$O$_{10}$N$_8$ 801.5444, found 801.5424, Δ$_{ppm}$ −2.07.

Demiguisin (**7**): white solid; $[\alpha]_D^{25}$ −18.7 (c 0.1 w/v%, MeOH); UV (acetonitrile), λ$_{max}$ 204 nm; IR (ATR) ν$_{max}$ 3731, 2938, 2539, 2364, 2024, 1690, 1550, 1020 cm$^{-1}$; NMR spectral data, see Supplementary Table 8; HRMS (ESI) *m/z* [M + H]$^+$ calcd. for C$_{30}$H$_{47}$O$_{10}$N$_9$ 694.3519, found 694.3520, Δ$_{ppm}$ 0.14.

## Biosynthetic pathway analysis

For phylogenetic reconstruction of condensation and ketosynthase domains reference sequences were retrieved from natural product domain seeker database (NaPDoS2_v13b)[46]. For further details, see Supplementary Methods.

## RNA sequencing and analysis

*Amycolatopsis* sp. PS_44_ISF1 pre-cultures were prepared in ISP2 broth and cultivated for seven days at 30 °C and shaking at 150 rpm. Then, agar plates (each *n* = 3) of a lipopeptide high-production (PDA) and low-production (CSA) medium were inoculated with 500 μl of the pre-culture. After cultivation for five days at 30 °C, mycelium of each plate was harvested, frozen in liquid nitrogen and stored at −80 °C until shipment. For RNA extraction, samples (*n* = 3 for CSA, and *n* = 3 for PDA) were pooled to obtain sufficient RNA amount and quality for sequencing. RNA extraction, sample quality control, library construction and RNA sequencing was performed by Novogene Co., Ltd (Cambridge, UK). Each set of reads (CSA, PDA) were mapped to the annotated genome of PS_44_ISF1 using Geneious Prime v2020.1.2 (Biomatters Ltd.). Expression levels depicted by normalized transcript counts (TPM = CDS read count * mean read length * 10$^6$)/(CDS length * total transcript count)[88] and differential expression confidence (DEC). DEC is the negative base 10 log of the *p*-value, adjusted to be negative for genes that are under-expressed in sample 2 (PDA) compared to sample 1 (CSA), or positive for over-expressed genes. The *p*-value corresponds to the probability that, for a given gene, a randomly selected transcript would come from that gene and is calculated as number of transcripts mapped to that gene/total number of transcripts from each sample. The probabilities for each sample are then multiplied to form the *p*-value. Afterwards, the expression data for selected core biosynthetic genes from BGC regions encoding for NRPS, PKS or NRPS/PKS hybrids was extracted and imported in RStudio v.1.4.1106 (R Foundation). The selected values BGCs and selected core biosynthetic genes with their expression values were summarized and the normalized TPM (log10 transformed) and DEC values were visualized in a heatmap created using pheatmap package v1.0.12 with color schemes generated by viridis package v0.6.2. For details, see Supplementary Figs. 76–78.

## Creation of *Amycolatopsis* sp. PS_44_ISF1 knockout mutants

For the construction of single-crossover knockout constructs the integrative *E. coli* – *Streptomyces* shuttle vector pKJ55 was digested with *Xba*I and *Sph*I (NEB, USA). A). Internal fragments (~2 kb) of NRPS or PKS candidate genes were amplified using a standard Phusion® High-Fidelity Polymerase protocol (NEB, USA). Standard PCR conditions were as follows: 98 °C/3 min, 35 cycles (98 °C/30 s, T$_a$ °C/45 s, 72 °C/60 s per kb), 72 °C/10 min. Due to the high GC-content of the genome, PCR reactions were performed in GC buffer supplemented with 5% DMSO (v/v) with primer annealing temperatures (T$_a$) 3–5 °C below primer melting temperature T$_m$. Primers contain a 20 bp overhang for subsequent cloning of the PCR product into the linearized pKJ55 vector backbone. Both, linearized pKJ55 and each PCR product were purified from 1% agarose gel using Zymoclean Gel DNA Recovery Kit (Zymo Research, USA). For cloning, Gibson Assembly (NEBBuilder® HiFi DNA Assembly Cloning Kit, NEB) was performed according to the manufacturer's protocol with a molar vector-insert ratio of 1:2 and incubated for 1 h at 50 °C. The reaction mixture was transformed in *E. coli* DH5α chemically competent cells and positive transformants were selected on LB agar with apramycin (50 μg/mL). All constructs were purified using innuPREP Plasmid Mini Kit (JenaAnalytik, Germany) and verified by sequencing using Seq_pKJ55_IF_fwd/ rev (Eurofins Genomics, Germany) (Supplementary Tables 14 und 15).

Knockout mutants of PS_44_ISF1 were generated via intergeneric conjugational transfer[54,55,89]. Methylation-deficient *E. coli* ET12567/ pUZ8002 was transformed with pKJ55-derived suicide vector via electroporation. For each vector construct three conjugations were prepared as follows: 10 mL ET12567 (pUZ8002/pKJ55-ko) in LB medium (supplemented with 50 μg/mL Apr, 25 μg/mL Kan, 25 μg/mL Cam) were grown at 37 °C to an OD$_{600}$ of 0.4–0.5, washed twice with ice-cold LB to remove residual antibiotics, and resuspended in fresh 1 mL LB without antibiotics. PS_44_ISF1 spore suspensions (50 μL, 3.5 ×10$^9$

spores/mL, $2 \times 10^8$ spores in 25% glycerol) were added to 500 μL of 2x YT medium and heat shocked at 50 °C for 10 min. Equal amounts (v/v) of donor and recipient cells were mixed and the bacteria were pelleted by centrifugation. The medium was discarded, and the pellets were resuspended in residual 100 μL of medium. The suspension was streaked on MS agar plates, supplemented with 10 mM $MgCl_2$ and varying concentrations of $CaCl_2$ (10, 25, 50 mM)[55,89]. After incubation for 16–20 h, the plates were overlaid with 1 mL of water containing 0.5 mg of nalidixic acid (NA) and 1.25 mg of apramycin, and incubation at 30 °C was continued for 5-7 days. Ex-conjugants were streaked on MS (with 50 μg/mL Apr and 25 μg/mL NA). For verification of knock out mutants, a PCR using primer pairs P1P2 and P1P3 was conducted. PCR products were then confirmed by Sanger sequencing (Eurofins Genomics, Germany), and production of metabolites verified by LC-HRMS/MS analysis (Supplementary Figs. 82–86). Confirmed mutants were stored as 50% glycerol stocks prior to further use.

### Tensioactivity tests

Isolated compounds were dissolved in water (final concentration 10 mM), of which 20 μL were placed on a hydrophobic surface (par-afilm 'M'). Tween-20 (10 mM) was used as tensioactive positive control. For better visualization, 0.0025% crystal violet was added to the droplet (Supplementary Fig. 88).

### Disc-diffusion based antimicrobial activity assays

SPE fractions and pure compounds (10 mg/mL in DMSO) were tested for their antimicrobial activity against ecologically relevant indicator strains, including feather-degrading bacteria *B. licheniformis* DSM13, *P. monteilii* DSM1388 and *K. rhizophila* DSM11926, and non-degrading feather isolates *B. thuringiensis* DSM104061 and *S. epidermidis* DSM103867, as well as the skin pathogenic yeast *C. albicans* SC5314. For more experimental details, see Supplementary Methods and Supplementary Tables 16–18.

### Protease inhibition assays

Pure compounds were dissolved in DMSO (final concentration 20 mM). The inhibitory effect of the compounds was determined for the following enzymes: rhodesain, cruzain, cathepsin L, cathepsin B, SARS-CoV2 $M^{pro}$, SARS-CoV $PL^{pro}$, DENV2 NS2B-NS3, ZIKV NS2B-NS3, urokinase (uPA), sortase A (SrtA). Fluorimetric assays were performed at a concentration of 200 μM and changes in fluorescence recorded over a period of 10 min in intervals of 30 seconds (exception: SrtA 30 min). For more experimental details, see Supplementary Methods and Supplementary Table 19[70–73,90–92].

### Reporting summary

Further information on research design is available in the Nature Portfolio Reporting Summary linked to this article.

### Data availability

Supplementary Information contains all details to experimental and analytical data. Analytical data (1D, 2D NMR) is available on Zenodo [https://doi.org/10.5281/zenodo.13125628]. The WGS data used in this study are available in the NCBI database under accession code JANUXO000000000.1/. The SRA data used in this study are available in the NCBI database under accession codes SRR21206835, SRR21206836 and SRR21206837. RNA sequencing data used in this study are available in the NCBI database under accession codes SRR21206833 and SRR21206834. The amplicon sequencing data generated in this study have been deposited in the NCBI database under accession codes SAMN30849808, SAMN30849809, SAMN30849810, SAMN30849811, SAMN30849812, SAMN30849813, SAMN30849814, SAMN30849815, SAMN30849816, SAMN30849817, SAMN30849818, SAMN30849819, SAMN30849820, SAMN30849821, SAMN30849822, SAMN30849823, SAMN30849824, SAMN30849825, SAMN30849826,

SAMN30849827 and SAMN30849828. We have uploaded the MS-Data to the MAssIVE Server under the accession code MSV000093302 [https://massive.ucsd.edu/ProteoSAFe/dataset.jsp?task=7c1ba2e849ed42f88592c69f6a599d83]. Source data are available in the Source Data file. Source data are provided with this paper.

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

## Acknowledgements

This study was funded by the German Research Foundation (DFG, Deutsche Forschungsgemeinschaft) under Project-ID 239748522 – CRC 1127 (project A6). We are grateful for the financial support from the Agence Nationale de la Recherche (ANR, ANR-17-CE07-0051-01), the German Research Foundation (DFG, BE 4799/3-1), the Carlsberg Foun-dation (Distinguished Associate Professor Fellowship, CF17-0248) and the Korean National Research Foundation (2021R1I1A1A0104960613 and 2018R1A6A1A03023718). We also would like to acknowledge support from the New Guinea Binatang Research Centre in Madang, the PNG National Museaum and Art Gallery in Port Moresby, the PNG Immigration & Citizen Service Authority and the Department of Environment and Conservation, Boroko, Papua New Guinea. We also would like to thank Daria Ripin (HKI) for technical assistance, Sven Balluff (HIPS) for con-ducting initial co-cultivation experiments, Christiane Weigel (HKI) for conducting antimicrobial assays, Heike Heinecke (HKI) for NMR mea-surements, Philipp Stephan (HKI) for helpful discussions and providing a Piz standard for Marfey's analysis, and Marie Dayras (HIPS) for the critical review of NMR and MS/MS data.

## Author contributions

E.S., S.U. and K.H.B. conceived the study. E.S., S.U., K.H.B., T.D., A.K., J.F., R.M., G.M., B.I. and H.M. performed the research and analyzed the data. E.S., S.U., K.H.B., T.D. and T.S. analyzed analytical data. K.A.J., G.M., M.P., B.I., T.S. and C.B. supervised the research and acquired funding. E.S., K.H.B., S.U. and C.B. wrote the paper with input from all co-authors. All authors contributed to the review and editing of the paper.

## Funding

## Competing interests

The authors declare no competing interests.

## Additional information

[1]Anti-infectives from Microbiota, Helmholtz-Institut für Pharmazeutische Forschung Saarland (HIPS), Campus E8.1, 66123 Saarbrücken, Germany. [2]Chemical Biology of Microbe-Host Interactions, Leibniz institute for Natural Product Research and Infection Biology – Hans-Knöll-Institute (HKI), Beutenbergstraße 11a, 07745 Jena, Germany. [3]Yonsei Institute of Pharmaceutical Sciences, College of Pharmacy Yonsei University, Songdogwahak-ro 85, Incheon 21983, Republic of Korea. [4]Natural History Museum of Denmark, Research and Collections University of Copenhagen, 2100 Copenhagen East, Denmark. [5]Department of Biomolecular Chemistry, Leibniz institute for Natural Product Research and Infection Biology – Hans-Knöll-Institute (HKI), Beutenbergstraße 11a, 07745 Jena, Germany. [6]Section for Ecology and Evolution, Department of Biology, University of Copenhagen, 2100 Copenhagen East, Denmark. [7]The New Guinea Binatang Research Centre, Madang, Papua New Guinea. [8]Papua New Guinea National Museum and Art Gallery, Port Moresby, Papua New Guinea. [9]Institute for Pharmaceutical and Biomedical Sciences (IPBW), Johannes Gutenberg University Mainz, Staudinger Weg 5, 55128 Mainz, Germany. [10]Swedish Museum of Natural History, Department of Bioinformatics and Genetics, P.O. Box 50007, SE-10405 Stockholm, Sweden. [11]Saarland University, Campus, 66123 Saarbrücken, Germany. [12]These authors contributed equally: Elena Seibel, Soohyun Um. ✉e-mail: Christine.beemelmanns@helmholtz-hips.de

