## [Peer Review File · Nature Communications]

REVIEWER COMMENTS

Reviewer #1 (Remarks to the Author):

Review of A toxic bird associated Amycolatopsis co-secretes two protective antimicrobial nonribosomally-synthesized peptide families

It will be a short review as the work is in the top tier in terms of quality. This is such a wonderful read and should be accepted nearly as is. So cool to see how commensals can inhibit feather damaging microbes.

Minor comments.

Fig 2E is really faint. I would brighten the data over the background a bit more.

Make the data public in GNPS and provide the actual networking task ID or the link to the job in the methods section. (I did not see it in the SI but perhaps just missed it).

Finally although it is up to the authors and editors what they want to have as title but a more general title would be way more fitting. Something aking to "Commensals from birds inhibit feather damaging bacteria"

Congrats on an amazing paper!

Reviewer #2 (Remarks to the Author):

This is an interesting report on the microbiome of *Pachycephala schlegelii* and the isolation of pachycephalamides from *Amycolatopsis*.

The chemical identifications are state of the art; also the identification of the gene clusters.

However, the antimicrobial analyses fall short of established methods in the field.

In microbiology, suspension assays are preferred over agar diffusion assays. Furthermore, quantitative data are required, not relative data. Thus, you need to provide MIC and MBC data for your compounds and antimicrobial activities. You should even consider the use of time-kill analyses to show if the activities are bacteriostatic or bacteriocidal.

The introduction needs to be more precise. Preen gland secretions have been studied from a number of birds. Apparently, there is no general scheme, which compounds and bacteria are present. Thus, you need to say, from which birds a finding came. Thus "parts of the identified communities include keratinolytic bacterial species, such as *Bacillus licheniformis* or *Kocuria rhizophila*" is too general.

Please consult more recent studies of preen glands by M Braun et al, who studied a broad diversity of birds (both in terms of waxes, and microbiomes).

You mentioned that you took samples from 5 birds and killed another 12 birds; did you find the same results in all individuals? Where there differences? Please provide details.

You mention several times, that the bird is a toxic species, but you need to explain the underlying toxicology in more detail (which is apparently not related to the preen gland chemistry).

As you know, if you collect biological material outside Europe, you have to follow the rules of the Nagoya convention; do you have MTA and other legal documents?

I wonder why you have not included a scientist from New Guinea as an author or coworker. You probably know that biopiracy is not allowed.

Reviewer #3 (Remarks to the Author):

This is a very interesting manuscript that describes a potential new bacterial species producer of two new nonribosomally peptide antibiotics. Their results are of high impact, not only for the discovery of new antimicrobials, but also for the ecological relevance of the bacterial producer in its host. In recent years have been an increasing interest in analyzing the role of avian microbial symbionts against infections in different avian species, and this paper broadens this body of knowledge. Overall, I think the authors present a good work with a big experimental and analytical effort and very interesting results. Nevertheless, I don't find a good agreement between the expressed goal and the results. The authors stated that they aim to illuminate the functions of microbial communities and their involvement in infection processes, but the paper mainly focused on the identification of *Amycolatopsis* sp. PS_44_ISF1 and characterization of the novel NRPs pachycephalamides and demiguisin (as is clearly shown in the title). For instance, according to the declared goal, beyond this strain, was there another antimicrobial producer in the culturable microbiota or was this the single one? What was the microbial composition of the culturable microbiota?

The main strength of the paper is the finding of this *Amycolatopsis* sp. at low abundance which produce detectable amounts of NRPs on feathers. In this sense, some deeper discussion should be given relative to the potential in vivo role given the low amounts of the producers and so the NRPs. I think this is still relevant, because despite the amounts the growth on a solid substrate (i.e. the feathers) might achieve local active concentrations, but some words to explain its potential role given the extremely low levels. On the other hand, the weakness of the papers is the biosynthetic pathways inference, which I find speculative.

Minor comments and questions:

In the microbiota composition old and new phyla nomenclature is mixed up, for instance Proteobacteria is used instead of Pseudomonadota, but Actinomycetota is used instead of Actinobacteria. Main databases for microbial massive identification still hold the old nomenclature, but the one used should be consistent.

On how many specimens of *P. schlegelii* was Amycolatopsis sp. found? This data should be given. In this sense, Do the specimens where compounds 1-7 were found fit? Can they be detected in the uropygial secretion despite their presence on feathers?

What does authors mean by “hybrid genome”?

The 16S similarities to close relatives should be given, as is the other gold standard in addition to DDH

The deletion mutants are a strong evidence of the function, but for a complete assignation complementation mutants would strengthen the assignment.

In the conclusions the authors state that “these NRPs represent the first examples of bioactive compounds isolated from a bacterial strain originating from an UG”. They should be more specific on what they mean by bioactive compounds, as they already show in their references some examples of antimicrobials from the UG.

Finally, the format of figures should be reviewed, there are repeated figures and misplaced figures in main manuscript and the supplementary material.

Point-to-point response

Reviewer #1	Congrats on an amazing paper!	We deeply appreciate the enthusiastic feedback and extend sincere gratitude for the reviewer's time and effort in evaluating our manuscript.
	Fig 2E is really faint. I would brighten the data over the background a bit more.	Many thanks, we have edited Figure 2
	Make the data public in GNPS and provide the actual networking task ID or the link to the job in the methods section. (I did not see it in the SI but perhaps just missed it).	We have uploaded the MS-Data to the MAssIVE Server: ID: MassIVE MSV000093302 Name: GNPS - Methanolic extracts of Amycolatopsis spp. on ISP2 - 8cae4ec6622e4cc0959042f087315336 We have uploaded the NMR to Zenodo. All data will be released prior to publication.
	Something aking to "Commensals from birds inhibit feather damaging bacteria"	We have rephrased the section.
Reviewer #2	The chemical identifications are state of the art; also the identification of the gene clusters.	We deeply appreciate the enthusiastic feedback and extend sincere gratitude for the reviewer's time and effort in evaluating our manuscript.
	The introduction needs to be more precise. Preen gland secretions have been studied from a number of birds.	We have indeed rephrased the section and included more references.
	Please consult more recent studies of preen glands by M Braun et al, who studied a broad diversity of birds (both in terms of waxes, and microbiomes).	Thank you for pointing out, we have included this study in our manuscript.
	Apparently, there is no general scheme, which compounds and bacteria are present. Thus, you need to say, from which birds a finding came. Thus "parts of the identified communities include keratinolytic bacterial species, such as Bacillus licheniformis or Kocuria rhizophila " is too general.	Thank you for pointing out. We have now included more detail in the text and also as figure in Figure 1, which showcases the relative, averaged composition. A more detailed depiction can be found in Table S1 and Figure S1, which shows the individual samples, locations and also details to each sample.
	You mentioned that you took samples from 5 birds and killed another 12 birds; did you find the same results in all individuals? Where there differences? Please provide details.	We apologize for not make the experimental set-up more clearer and have now included the sample numbers in the text. We have evaluated the bacterial alpha/beta diversities. For details, see Supporting Information Chapter 5. However, we didn't detect an effect of the interaction between the sample type and sample region

		(PERMANOVA: $F_{df} = 1.048_1$, $R^2 = 0.0393$, $p = 0.3503$), indicating that microbiome compositions are consistent among P. schlegelii individuals across different geographic regions. However, the composition of P. schlegelii UG microbiomes differed notably from temperate bird species.
	You mention several times, that the bird is a toxic species, but you need to explain the underlying toxicology in more detail (which is apparently not related to the preen gland chemistry).	We apologize for missing this detail. the New Guinean toxic bird Pachycephala schlegelii (Regent Whistler) is known to integrate the neurotoxic steroid alkaloid batrachotoxin (BTX) into feathers likely to deter predators. Feather samples extracts also do contain BTX. However, as the presence of this toxin is food and season dependent and thus does likely not depend on the presence/absence of symbionts. However, we acknowledge that the direct correlation/causation remains yet to be determined.
	However, the antimicrobial analyses fall short of established methods in the field. In microbiology, suspension assays are preferred over agar diffusion assays. Furthermore, quantitative data are required, not relative data. Thus, you need to provide MIC and MBC data for your compounds and antimicrobial activities. You should even consider the use of time-kill analyses to show if the activities are bacteriostatic or bacteriocidal.	We indeed agree that both suspension assays (broth dilution assays) as well as disc diffusion assays are commonly used to determine the growth inhibitory effects for bacteria. Disc diffusion assays have the advantage that it allows to visualize if resistant colonies re-emerge, which cannot be differentiated in a broth dilution assay. Furthermore, in case of filamentous microorganisms, broth dilution assays are less commonly used due to the unreproducible read-out. In this manuscript, we have depicted the plate assays for visualization of the activity. In addition, we do not elaborate on the strength of the activity in terms of numbers (quantitatively, MIC, MBC) as the new compounds are only moderately active with estimated MICs above 12.5 $\mu\text{g}/\text{mL}$ (calculated from concentration used in different disc diffusion assays). While we certainly can quantify in terms of MICs, the overall statement in terms of moderate effects will remain the same. Here, we would also like to highlight that we have rephrased this and other sections to stress the fact that the secretion of a cocktail of compounds is likely more important for the birds protection. This is because during the revision process, we realized that the co-eluting macrolactam ciromicin A, is likely mainly responsible for the antifungal activity and might have contaminated previous NMR-pure samples. However, here it needs to be noted that NMR-purity appears to be not sufficient as the remaining contaminates (<5%) are accountable for the observed antifungal activity, as previously reported by us and others (References: 29 and 30). We deeply apologize for this insufficient quality control, as this should have not happened. Thus, we are very thankful for the chance to revise our manuscript.

		While demiguisin is not responsible for the observed antifungal effect, the co-produced macrolactam, rifamycin congeners, and pachycephalamides collectively form the antimicrobial cocktail responsible for the observed activity in co-culture and assays. This cocktail effectively inhibits both bacteria and yeast, and thus likely as protective agents. In case of the protease inhibition assay, the data has been quantified due to the nature of the assay.
	As you know, if you collect biological material outside Europe, you have to follow the rules of the Nagoya convention; do you have MTA and other legal documents? I wonder why you have not included a scientist from New Guinea as an author or coworker. You probably know that biopiracy is not allowed.	Indeed, we have a long-standing collaboration with researchers across the globe and are well aware that “biopiracy” is not allowed now tolerated by the community, or legislative apparatus. Here, it should be mentioned that the country of origin has not signed the Nagoya convention. https://www.cbd.int/abs/nagoya-protocol/signatories Regardless of this legal framework, the collections were done with permission as stated in the supporting information (Table S1) in the submission. We agree that we should not only acknowledge the institutions that supported the study but also integrate them more intensively into the publication process. Thus, we are more than happy to have included two of the researcher that coordinated the permit process and local activities as authors.
Reviewer #3 (Remarks to the Author):	This is a very interesting manuscript that describes a potential new bacterial species producer of two new nonribosomally peptide antibiotics. Their results are of high impact, not only for the discovery of new antimicrobials, but also for the ecological relevance of the bacterial producer in its host. Overall, I think the authors present a good work with a big experimental and analytical effort and very interesting results.	We greatly appreciate the constructive feedback and the time invested by the reviewer in thoroughly evaluating our manuscript.
	Nevertheless, I don't find a good agreement between the expressed goal and the results. The authors stated that they aim to illuminate the functions of microbial communities and their involvement in infection processes, but the paper mainly focused on the identification of Amycolatopsis sp. PS_44_ISF1 and characterization of the novel NRPs pachycephalamides and demiguisin (as is clearly shown in the title).	We are thankful for this constructive comment and have also rephrased the introduction and discussion accordingly. Indeed, we intentionally decided to more focus on the isolate compared to broader screening of isolates as we aimed to explore in addition the biochemistry and natural product repertoire for the first time more in detail. We indeed isolated several bacterial candidates showing either antibacterial or antifungal activity, but neither of them yet showed both bioactivities.

	For instance, according to the declared goal, beyond this strain, was there another antimicrobial producer in the culturable microbiota or was this the single one? What was the microbial composition of the culturable microbiota?	We acknowledge that analysis of other isolates – also those belonging to more abundant bacterial families – is certainly a very important topic and is currently pursued in the laboratory. However, elaboration on the chemistry and biochemistry of even more isolates would be beyond the scope of this manuscript. As in all cases, the culturable microbiota diverges from those observed in amplicon sequencing data. However, we were yet able to isolate representatives of several detectable families. We have addressed the topic also now more in detail in the first result section.
	The main strength of the paper is the finding of this Amycolatopsis sp. at low abundance which produce detectable amounts of NRPs on feathers. In this sense, some deeper discussion should be given relative to the potential in vivo role given the low amounts of the producers and so the NRPs. I think this is still relevant, because despite the amounts the growth on a solid substrate (i.e. the feathers) might achieve local active concentrations, but some words to explain its potential role given the extremely low levels.	We are thankful for the comment and understand that it would be very interesting to speculate on the abundance and local concentrations of ecological relevance. We have indeed address this question by pursuing MALDI Imaging experiments of co-cultures, which clearly show the abundances of signals in proximity to the co-occurring microorganism, which also allow to relatively assign concentration ranges to each m/z feature. However, when considering the natural growth habitat, we yet do not know the production titers of natural products of producers and thus it remains rather speculative, which local concentrations are important. In an effort to address the question, we also tested MALDI Imaging of feathers, but yet failed to depict any exogenous natural products on the surface or within the feathers. Thus, spatial resolution analysis is yet another technical hurdle to overcome. In addition, we have only taken breast feathers for analysis to not further harm birds (that were not scarified for UG analysis) more than necessary, which limits comparative studies. Here it should also be noted that Pseudonocardiaaceae, as discussed in the first section, accounted for 1.5% of the feather, and 0.14% of the UGs microbiome. Amplicon sequence variants assigned to Amycolatopsis were detectable in nearly half of all collected samples. Thus, it is important to conclude that Amycolatopsis sp. PS_44_ISF1 is not an obligate symbiont (mutualist), but that the presence of this genus as well as members of the family are likely belonging to the protective microbiome the birds require for their health care.
	In the microbiota composition old and new phyla nomenclature is mixed up, for instance, Proteobacteria is used instead of Pseunomonadota, but Actinomycetota is used instead of	We apologize for this mistake and have corrected the nomenclature and also run the analysis using an updated software and database.

	Actinobacteria. Main databases for microbial massive identification still hold the old nomenclature, but the one used should be consistent.	
	On how many specimens of P. schlegelii was Amycolatopsis sp. found? This data should be given. In this sense, Do the specimens where compounds 1-7 were found fit? Can they be detected in the uropygial secretion despite their presence on feathers?	We have included this information now in Figure S1 in more detail! As stated now in chapter 5: Among the Actinobacteriota, Pseudonocardaceae accounted for only 1.5% of bacterial sequences in feather microbiomes, but only 0.14% in UG (Figure 1D-E). We detected 16 ASVs belonging to the genus Amycolatopsis , where seven out of nine feather samples and three out of 13 UG samples harbored Amycolatopsis . While, we were unable to detect Amycolatopsis sequences in the UG of NHMD 615979 (the individual we isolated the Amycolatopsis strain from), two AVS belonging to Amycolatopsis were detected in the feather sample of the individual (Figure S1E).
	The 16S similarities to close relatives should be given, as is the other gold standard in addition to DDH	We have performed whole genome-based taxonomic analysis and the genome is deposited at NCBI Phylogenetic analysis was indeed also done using the 16 rDNA sequence as stated in chapter 7. [...] 16S rDNA sequences were extracted from the user genomes using RNAmmer and subsequently BLASTed against 16S rDNA of all type strains available in the TYGS database....[] The length of the retrieved 16 rDNA sequence is depicted in Figure S2 in comparison to other sequences of type strains used in this study.
	On the other hand, the weakness of the papers is the biosynthetic pathways inference, which I find speculative. What does authors mean by “hybrid genome”?	Hybrid genome assembly is described in chapter 7 and is nowadays the best way to most accurately obtain whole genome information compared to short or only long-read sequencing. We have now referred to it as: “To enable a more accurate phylogenomic placement of Amycolatopsis sp. PS_44_ISF1, its hybrid genome assembly was created by merging paired-end shotgun sequencing with long-read sequencing from PacBio and Nanopore technologies”
	The deletion mutants are a strong evidence of the function, but for a complete assignation complementation mutants would strengthen the assignment.	We agree that complementation of deletion mutants is a commonly employed method for verifying functionality of single gene deletion mutants to restore certain phenotype observations. However, when dealing with deletion mutants targeting gene sequences encoding hybrid megaenzymes comprised of multiple genes, complementation approaches are not widely utilized due to their technical susceptibility to failure. Complementation strategies are impractical for deletion mutants of large non-canonical NRPS variants, as the considerable size of the deleted

		sequences poses challenges to gene synthesis or cloning. Even if synthesis or amplification of such large sequences with often repetitive sequences is possible, introducing them on a plasmid for correct transcription, translation, and functional arrangement within the wild-type host remains very uncertain. While the deleted genes are integral components of complete BGCs, they only encode a portion of the full NRPS arrangement. Thus, it remains speculative whether they would function effectively when introduced independently, and with an imbalanced protein ratio/production. Moreover, it's worth noting that we have generated a total of fourteen distinct mutants targeting various PKS and NRPS regions. AS discussed in the text additional established measures (prediction softwares etc) were taken into consideration to exclude other gene candidates. These mutants were validated through sequencing and LC-MS/MS-based metabolomics, comparing metabolite levels to those of the wild type, where the presence or absence of confirmed metabolic traits serves as a metric
	In the conclusions the authors state that “these NRPs represent the first examples of bioactive compounds isolated from a bacterial strain originating from an UG”. They should be more specific on what they mean by bioactive compounds, as they already show in their references some examples of antimicrobials from the UG.	We have rephrased this section. Thank you for pointing out.
	Finally, the format of figures should be reviewed, there are repeated figures and misplaced figures in main manuscript and the supplementary material.	We have revised all manuscripts and figures in manuscript and SI. While some figures or text passages might appear to be repetitive, we would kindly consider these figures also in the context of the very extensive supplement documents and should help to assist the reader in understanding and following the supporting information as a standalone document. However, if requested, we are happy to remove any redundancy within the supporting information.

REVIEWERS' COMMENTS

Reviewer #1 (Remarks to the Author):

Sorry it took me so long to provide the re-review. Got backlogged with end-of-school year duties and major grants. The editors did a great job in following up.

I already really enjoyed this paper and it only got better. The additional clarifications really helped the manuscript.

I did check the access to the MS data in GNPS/MassIVE but i cannot find it. Please make it public. Once this is public the paper should be accepted.

Congrats on a nice paper.

Reviewer #2 (Remarks to the Author):

The authors have addressed most concerns and recommendations of my first review.

However, the most serious criticism has been discussed only.

The methodology for antimicrobial testing using the disc diffusion assay is outdated.

Such a paper in Nature would require state of the art testing, such as broth dilution assays and time kill assays.

You need MIC and MBC values if you want to compare and discuss antimicrobial activities

Reviewer #3 (Remarks to the Author):

Thank you for the revised manuscript. I think that the new version has been improved. Nevertheless, I'd have appreciated to specify the specific lines of the corrections to speed up the review.

I think that most of reviewers' comments have been addressed. Nevertheless, I still see a mixture old and new phyla nomenclature. The authors state that they "have corrected the nomenclature and also run the analysis using an updated software and database", but in the results section still appears Proteobacteria (instead of Pseudomonadota) and Actinomycetota (instead of Actinobacteria). I don't see a big issue using old nomenclature, specifically because, as seen in supplementary dataset, this is the one that have been used for identification. But I think the authors should be consistent and do not mix different taxonomic schemes.

Point-to-point response to reviewer requests

Reviewer #1 (Remarks to the Author):	
I already really enjoyed this paper and it only got better. The additional clarifications really helped the manuscript.	Many thanks, the positive answer is very much appreciated
I did check the access to the MS data in GNPS/MassIVE but i cannot find it. Please make it public. Once this is public the paper should be accepted.	We have changed the release data and apologies for not noting this earlier. The hyperlink has been added to the data availability statement
Reviewer #2 (Remarks to the Author): The authors have addressed most concerns and recommendations of my first review. However, the most serious criticism has been discussed only. The methodology for antimicrobial testing using the disc diffusion assay is outdated. Such a paper in Nature would require state of the art testing, such as broth dilution assays and time kill assays. You need MIC and MBC values if you want to compare and discuss antimicrobial activities.	Many thanks for reading the article carefully. We acknowledge the primary concern raised by the reviewer and agree that while diffusion assays are useful for preliminary screening, they should be replaced by MIC, IC50, or MBC values when appropriate. For the newly reported compound families discussed, we focus on the prescreening and their tensioactive properties. As detailed in the text (result and discussion), the activity spectrum of individual compounds was measurable, but in some cases, they were below the threshold typically used for determining IC50, MBC, or MIC values. Due to the moderate antimicrobial properties likely resulting from their amphiphilic nature, measuring MICs is technically feasible but not pharmacologically or ecologically relevant. For the two annotated features (macrolactams and rifamorpholines), MIC and IC50 values have been reported and cited in the manuscript. Additionally, we have omitted a detailed comparison of activity ranges for these specific reasons.
Reviewer #3 (Remarks to the Author):	
Nevertheless, I still see a mixture old and new phyla nomenclature. The authors state that they “have corrected the nomenclature and also run the analysis using an updated software and database”, but in the results section still appears Proteobacteria (instead of Pseudomonadota) and Actinomycetota (instead of Actinobacteria). I don’t see a big issue using old nomenclature, specifically because, as seen in supplementary dataset, this is the one that have been used for identification. But I think the authors should be consistent and do not mix different taxonomic schemes.	We apologize for not being consistent and the reviewer is correct that we still mixed up the nomenclature in the main manuscript of the revision. For the Supporting Information, it might have happened that accidentally the old Supplementary file 1 was reviewed. We have now made sure that the new nomenclature in the reanalyzed data has been attached to this this revision. The data has been uploaded to NCBI under the number: SAMN30849808-SAMN30849828

<https://massive.ucsd.edu/ProteoSAFe/dataset.jsp?task=7c1ba2e849ed42f88592c69f6a599d83>

MassIVE Dataset Information	
Title	GNPS - Methanolic extracts of Amycolatopsis spp. on ISP2 - 8cae4ec6622e4cc0959042f087315336
Description	Comparative metabolomics analysis of three Amycolatopsis strains isolated from distant geographical locations.
MassIVE Accession	MSV000093302
Dataset Type	Public Partial
Principal Investigators	Christine Beemelmanns (christine.beemelmanns@helmholtz-hips.de), Helmholtz Institute for Pharmaceutical Research Saarland, Germany
Username	ka26lu
Contact Email	elena.seibel@hki-jena.de
Species	Amycolatopsis saalfeldensis (NCBITaxon:394193) Amycolatopsis sp. M39 (NCBITaxon:1825094) Amycolatopsis sp. PS_44_ISF1
Instrument	MS:1002634
Post-Translational Modifications	MS:1002864
Keywords	Amycolatopsis Comparative metabolomics Antimicrobial compounds ecology-guided natural product isolation
Number of Files	17
Total Size	649.98 MB
Subscribers	1
Subscription Status	Unsubscribe
Analyze Data	Analyze Submitted Spectra Import and Analyze Dataset with Networking Now!
FTP Download	ftp://massive.ucsd.edu/MSV000093302/
Browse Visualizable Dataset Files	View Dataset Files in Browser
Other Dataset Actions	Add Files Add/Update Metadata Add Publication
Add Comment	Comment on Dataset Attach Reanalysis Results
Publications